# Application of copper(II)-based chemicals induces CH₃Br and CH₃Cl emissions from soil and seawater

Yi Jiao [1,4 ✉], Wanying Zhang [2], Jae Yun Robin Kim[1], Malte Julian Deventer[1,5], Julien Vollering [1,6] & Robert C. Rhew [1,3 ✉]

Methyl bromide ($CH_3Br$) and methyl chloride ($CH_3Cl$) are major carriers of atmospheric bromine and chlorine, respectively, which can catalyze stratospheric ozone depletion. However, in our current understanding, there are missing sources associated with these two species. Here we investigate the effect of copper(II) on $CH_3Br$ and $CH_3Cl$ production from soil, seawater and model organic compounds: catechol (benzene-1,2-diol) and guaiacol (2-methoxyphenol). We show that copper sulfate ($CuSO_4$) enhances $CH_3Br$ and $CH_3Cl$ production from soil and seawater, and it may be further amplified in conjunction with hydrogen peroxide ($H_2O_2$) or solar radiation. This represents an abiotic production pathway of $CH_3Br$ and $CH_3Cl$ perturbed by anthropogenic application of copper(II)-based chemicals. Hence, we suggest that the widespread application of copper(II) pesticides in agriculture and the discharge of anthropogenic copper(II) to the oceans may account for part of the missing sources of $CH_3Br$ and $CH_3Cl$, and thereby contribute to stratospheric halogen load.

[1] Department of Geography, University of California, Berkeley, CA 94720, USA. [2] Anhui Province Key Laboratory of Polar Environment and Global Change, School of Earth and Space Sciences, University of Science and Technology of China, 230026 Hefei, China. [3] Department of Environmental Science, Policy and Management, University of California, Berkeley, CA 94720, USA. [4]Present address: Terrestrial Ecology Section, Department of Biology, University of Copenhagen, Universitetsparken 15, Copenhagen Ø DK-2100, Denmark. [5]Present address: ANECO Institut für Umweltschutz GmbH & Co., Hamburg 21079, Germany. [6]Present address: Department of Environmental Sciences, Western Norway University of Applied Sciences, 5020 Bergen, Norway. ✉email: yi.jiao@bio.ku.dk; rrhew@berkeley.edu

Methyl bromide ($CH_3Br$) and methyl chloride ($CH_3Cl$) are the most abundant brominated and chlorinated hydrocarbons in the atmosphere[1]. They have average tropospheric lifetimes of 0.8 and 0.9 years[1], respectively, which are long enough for them to reach the lower stratosphere and disassociate, releasing bromine and chlorine radicals that catalyze stratospheric ozone depletion. However, the sizes of identified sources of $CH_3Br$ and $CH_3Cl$ are smaller than those of the sinks with a $-39$ Gg yr$^{-1}$ imbalance for $CH_3Br$ and a $-748$ Gg yr$^{-1}$ imbalance for $CH_3Cl$, respectively[1]. A recent reevaluation on $CH_3Cl$ emissions from tropical plants[2], which was seen as the largest $CH_3Cl$ source, showed it may be overestimated by about 1300 Gg yr$^{-1}$, suggesting the existence of even larger missing $CH_3Cl$ sources than previously thought. These discrepancies need to be resolved to better understand stratospheric halogen chemistry and to predict future ozone recovery. The present atmospheric $CH_3Br$ and $CH_3Cl$ originate from predominantly natural sources. These natural sources tend to be amplified under anthropogenic-induced influences, such as biomass burning[3], global warming[4,5], sea level rise[6,7], agricultural cultivation[8], land use change[9], etc., posing persistent and growing challenges for the recovery of stratospheric ozone[10,11].

Among the known sources and sinks of $CH_3Br$ and $CH_3Cl$, soil is an important bi-directional interface with simultaneous abiotic production[12,13] and biotic degradation[14,15]. Within the soil, abiotic Fenton-like reactions that involve Fe(III) oxidizing organic matter in the presence of halide ions are responsible for the production of methyl halides[12]. This pathway is ubiquitous and makes a significant contribution to the global methyl halide budgets. It is reasonable to postulate that the Fenton-like mechanism may be catalyzed by other transition metal ions, such as copper(II).

A setting that copper may be present in abundance is in global agriculture, where copper(II)-based pesticides, herbicides, and fungicides, such as copper(II) sulfate ($CuSO_4$), are widely used, especially in the fields of rice, oranges, walnuts, and grapes[16]. For example, in 2008 in the U.S., copper compounds accounted for 17% (the second most) of total fungicides applied in agriculture[17]. Roughly $2.0 \times 10^{11}$ kg $CuSO_4$ per year are used in global agriculture (Supplementary Note 1), in addition to other copper products.

Here, we incubate agricultural soil, seawater, and model compounds of soil organic carbon with different copper treatments, and quantify $CH_3Br$ and $CH_3Cl$ production rates, in order to explore and evaluate the potential effect of copper(II) chemicals on $CH_3Br$ and $CH_3Cl$ production. The results provide insight on copper-catalyzed production mechanisms of $CH_3Br$ and $CH_3Cl$, as well as the potential impact of copper chemical application on atmospheric $CH_3Br$ and $CH_3Cl$ budgets.

## Results and discussion

**$CH_3Br$ and $CH_3Cl$ flux from soil**. Live, unamended agricultural soil samples were net sinks of methyl bromide ($CH_3Br$) and methyl chloride ($CH_3Cl$) with average fluxes at $-0.06$ ng kg$^{-1}$ hr$^{-1}$ and $-3.7$ ng kg$^{-1}$ hr$^{-1}$, respectively (Fig. 1). After being autoclaved, they switched to $CH_3Br$ and $CH_3Cl$ sources at 0.45 ng kg$^{-1}$ hr$^{-1}$ and 6.9 ng kg$^{-1}$ hr$^{-1}$, respectively, supporting the conventional idea that soils act as a bi-directional interface of $CH_3Br$ and $CH_3Cl$ with simultaneous enzymatic degradation and abiotic production[5,12,18]. The deactivation of methyl halide-consuming microorganisms following heat treatment[19] in autoclaved soil results in the unidirectional interface of $CH_3Cl$ and $CH_3Br$ emissions, possibly catalyzed by endogenous Fe(III) in the soil[12]. $CH_3Br$ and $CH_3Cl$ fluxes declined to near zero after organic matter depletion, demonstrating that soil organic matter provides the carbon substrate for $CH_3Br$ and $CH_3Cl$ formation rather than atmospheric $CH_4$ or $CO_2$.

**Cu(II) amendment induces the production of $CH_3Br$ and $CH_3Cl$**. Addition of copper(II) sulfate ($CuSO_4$) at different dosages yielded enhancements in $CH_3Br$ and $CH_3Cl$ productions from both live and autoclaved soil samples, and the increments were positively correlated to the amount of $CuSO_4$ added (Fig. 1). No such trend was observed from the organic matter-depleted mineral soil. It has been previously demonstrated that halide ions can be alkylated to produce volatile halogenated carbons when soil or sediment organic matter is oxidized by Fe(III) without the mediation of sunlight or microbial activities[12]. The results suggested that Cu(II), as an electron acceptor, may oxidize organic matter in a similar manner as Fe(III) during the halogenation process to produce volatile $CH_3Br$ and $CH_3Cl$.

Notably, live soils produced 4–6 times more $CH_3Cl$ than $CH_3Br$ on molar basis (e.g., amended with 10 ml of 5 mM or 10 mM $CuSO_4$), despite Cl$^-$ being ~135 times more abundant than Br$^-$ in the soils (Supplementary Table 1). For autoclaved soil, $CH_3Br$ production rate was even up to 15 times that of $CH_3Cl$ (e.g., amended with 10 ml of 5 mM $CuSO_4$). These results suggested that Cu(II)-catalyzed methylation reaction favors bromide over chloride when both ions are present, possibly due to relative electronegativities and polarizabilities of the halogens[20]. This is a similar trend as found in the Fe(III)-catalyzed production of methyl halides[12,13,21].

The increases of $CH_3Br$ and $CH_3Cl$ production rates induced by copper(II)-addition were much larger in autoclaved soil samples than in live soil samples. For example, as the Cu(II) addition increased from 0 to 10 mM, average $CH_3Br$ fluxes from live soil samples gradually switched from $-0.06$ to 0.69 ng kg$^{-1}$ hr$^{-1}$, while emissions from autoclaved soils increased from 0.45 to 578 ng kg$^{-1}$ hr$^{-1}$. For $CH_3Cl$ fluxes, they shifted from $-3.7$ to 2.1 ng kg$^{-1}$ hr$^{-1}$, and from 6.9 to 25.2 ng kg$^{-1}$ hr$^{-1}$, respectively. It is possible that thermal treatment promoted the fracture of labile humic substances in soil[22–24], making the organic carbon more readily available for the formation of volatile halogenated compounds.

**In conjunction of Cu(II), $H_2O_2$ further promotes $CH_3Br$ and $CH_3Cl$ production**. The addition of $H_2O_2$ also enhanced $CH_3Br$ and $CH_3Cl$ production in both live and autoclaved soils (Fig. 2), but not as large as those caused by the addition of Cu(II). However, $H_2O_2$ amplified the effect of Cu(II) on $CH_3Cl$ and $CH_3Br$ production in both live and autoclaved soil samples (Fig. 2). Compared to experiments with Cu(II) addition only, the amendment of both $H_2O_2$ and Cu(II) increased the production rates of $CH_3Br$ and $CH_3Cl$ by about 93-fold and 7-fold, respectively, in live soils; and by about 1.2-fold and 5-fold, respectively, in autoclaved soils. It is postulated that the combination of Cu(II) and $H_2O_2$ produced hydroxyl radicals ($\bullet$OH)[25], which provides a powerful, nonspecific oxidant agent. Subsequently, the $\bullet$OH radicals react with organic compounds, halide ions, and/or copper ions to enhance the copper-mediated reaction to form volatile organochlorine compounds, such as $CH_3Br$ and $CH_3Cl$.

Catechol (benzene-1,2-diol) is regarded as a precursor of soil humic substances, and it is often used as a model substance of monomeric phenols that are commonly formed during the microbial degradation of many naturally occurring and anthropogenic aromatic substances[26]. To elucidate abiotic chemical reactions, in the absence of soil microbes or complex soil organic matter in soils that may be chemically altered during the autoclaving process, an experiment was designed with catechol to further test the essentiality of Cu(II) and $H_2O_2$ in $CH_3Br$ and $CH_3Cl$ production.

Water solutions containing catechol (10 mM), KBr or KCl (20 mM), $CuSO_4$ (50 mM), and $H_2O_2$ (50 mM) produced large

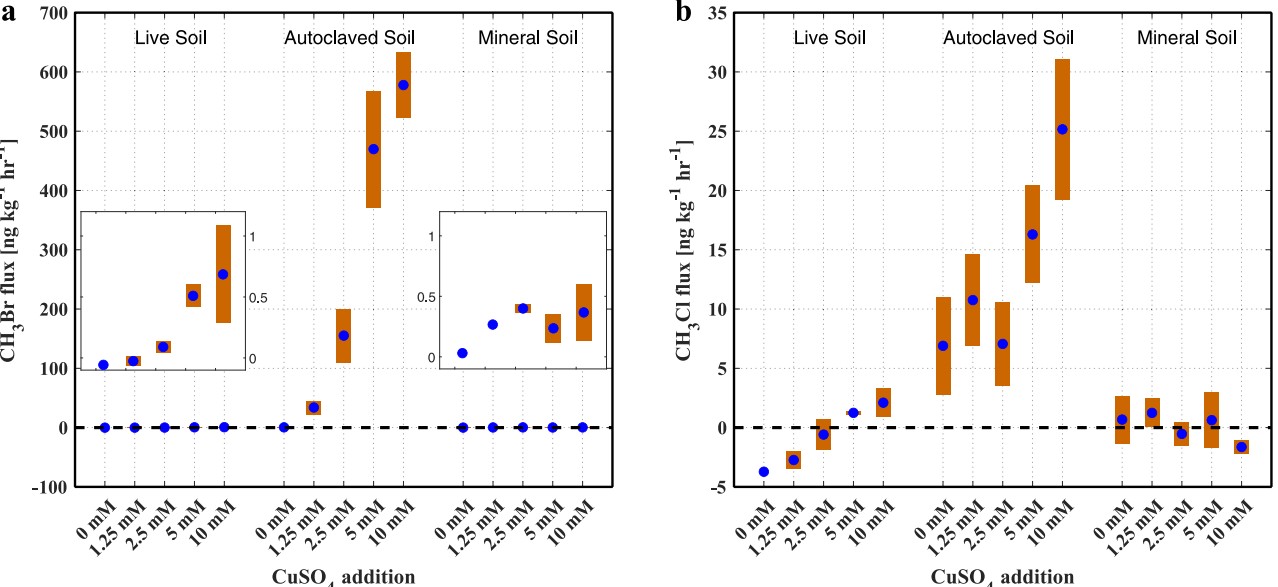

**Fig. 1 Methyl halide fluxes from soil samples amended with different amount of CuSO₄. a** Methyl bromide ($CH_3Br$) and **b** methyl chloride ($CH_3Cl$) fluxes from live, autoclaved, and mineral soil samples mixed with 10 ml of different concentrations of $CuSO_4$ solution. The blue dots represent the average fluxes of two replicate incubations ($n = 2$) and the upper and lower boundary of the bars represent the two individual fluxes. The inserted plots in (**a**) are the enlarged view of $CH_3Br$ fluxes from live and mineral soils.

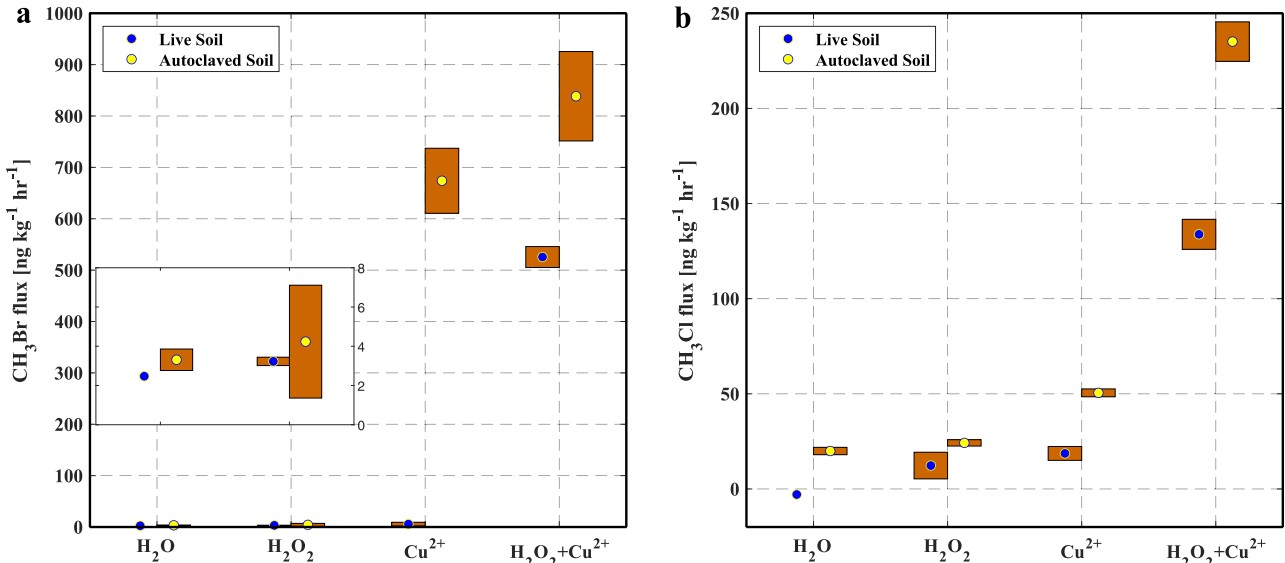

**Fig. 2 Methyl halide fluxes from soil samples amended with different chemicals. a** Methyl bromide ($CH_3Br$) and **b** methyl chloride ($CH_3Cl$) fluxes from live and autoclaved soil samples under different treatments. (i) 20 ml of deionized water was added; (ii) 10 ml each of deionized water and 50 mM $H_2O_2$ was added; (iii) 10 ml each of deionized water and 50 mM $CuSO_4$ was added; (iv) 10 ml each of 50 mM $H_2O_2$ and 50 mM $CuSO_4$ at were added. The dots represent the average fluxes of two replicated incubations ($n = 2$) in each treatment and the upper and lower boundary of the bars represent the two individual fluxes. The inserted plots in (**a**) are the enlarged view of $CH_3Br$ fluxes from soils in the first two treatments.

amounts of $CH_3Br$ and $CH_3Cl$, respectively (Supplementary Fig. 1), associated with an abrupt color change from clear to brownish with visible precipitates (Supplementary Fig. 2). $CH_3Br$ and $CH_3Cl$ production rates also increase with increasing concentrations of $CuSO_4$ and $H_2O_2$, which further demonstrate the enhancing effect of Cu(II) and $H_2O_2$.

**Time and sunlight effect**. The experiments described so far showed that the addition of $CuSO_4$ induces production of $CH_3Br$ and $CH_3Cl$ from the reaction with soil organic carbon.

In addition, Cu(II) works in conjunction with $H_2O_2$, an oxidizing agent, to further increase $CH_3Br$ and $CH_3Cl$ production. In natural ecosystems, the formation of $H_2O_2$ results principally from exciting humic substances by solar radiation[27,28]. Therefore, time-series light *vs.* dark experiments were conducted with soil samples to explore how long the Cu(II) effect can persist, and to find out if a potential photochemical pathway involving ambient sunlight exists.

The time-series experiments showed that, in the dark experiments (Fig. 3, dark incubations), the production rates of $CH_3Br$ and $CH_3Cl$ were highest immediately following the addition of $CuSO_4$,

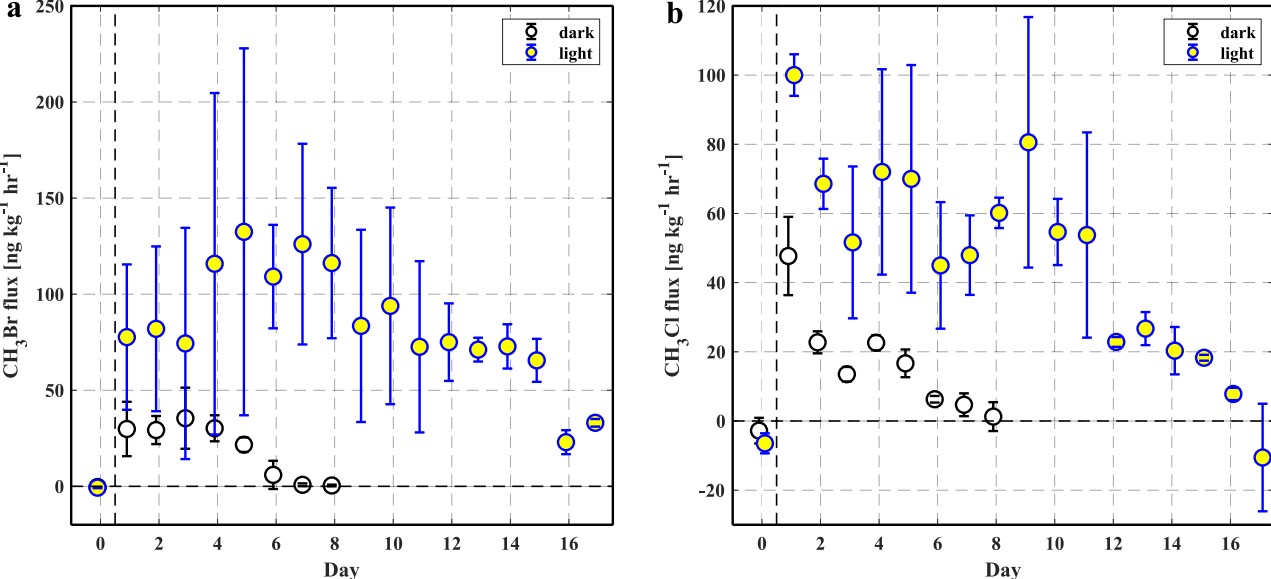

**Fig. 3 Time-series fluxes of methyl halide from soil samples. a** Methyl bromide ($CH_3Br$) and **b** methyl chloride ($CH_3Cl$) from light *vs.* dark incubations of live soil samples. On day 1, 10 ml of 50 mM $CuSO_4$ was added to soil samples (50 grams); the dark incubation lasted for 8 days while the light incubation was 17 days. The circles represent the average fluxes of triplicate incubations ($n = 3$) and the error bars represent the standard deviations.

and then declined to negligible levels over the following week. The depletion of Cu(II) may contribute to the persistent decline of $CH_3Br$ and $CH_3Cl$ production. Therefore, under natural conditions where copper can undergo aerial oxidation[20], it may apply a longer enhancing effect on $CH_3Br$ and $CH_3Cl$ production. It is noted that $CH_3Cl$ fluxes started to turn negative on day 8, indicating net consumption of $CH_3Cl$ from the ambient air, which might be attributable to the re-habitation of methyl halide-consuming microbial communities[27] after the addition of Cu(II) chemical.

For sunlight experiments (Fig. 3), the average $CH_3Br$ and $CH_3Cl$ fluxes were persistently higher than those from the dark experiments. Although the decline of $CH_3Br$ and $CH_3Cl$ fluxes over time was also observed, it took a longer period (17 days *vs.* 8 days) for them to resume near-background levels. These results suggested that solar radiation may amplify the effect of Cu(II) on $CH_3Br$ and $CH_3Cl$ production, and Cu(II) might be recycled under the light conditions to exert the longer enhancement effect.

Sunlit incubations of catechol were also conducted to further test the essential role of solar radiation played in $CH_3Cl$ production. Results showed rapid production rates occurred when the chemical solutions with Cu(II) were exposed to sunlight, while the omission of sunlight yielded slow production of $CH_3Cl$ (Supplementary Fig. 3a). It indicates the likely existence of a photochemical production pathway, in which copper may be reoxidized to exert a persistent enhancement effect on methyl halide production.

A few studies have proven the importance of irradiation in the production of methyl halides[29–32]. For example, artificial ultraviolet irradiation ($\lambda = 254$ nm) can catalyze the reaction between dissolved organic carbon and chlorine in seawater to generate $CH_3Cl$[32]. However, in natural environments, UV radiation with such a short wavelength would have been absorbed by stratospheric ozone before it can reach earth's surface, which makes this pathway only plausible in sewage treatment industries. Similar to some other studies[29–31], the light involved in our experiments was ambient sunlight. It has been proposed that the UVB region of sunlight may bring the complex of Cu(II) and halide ions to a redox excited state[20], promoting methyl halide production. Therefore, the larger $CH_3Br$ and $CH_3Cl$ fluxes in samples with copper sulfate and sunlight imply that sunlight is an important driver for the reaction.

On the other hand, the glass jars used in the sunlight experiments attenuated UVB ($\lambda = 265$–322 nm), UVA ($\lambda = \sim320$–390 nm), and photosynthetic active radiation (PAR, $\lambda = 410$–655 nm) by about 75%, <1%, and 23%, respectively (Supplementary Table 3), suggesting that $CH_3Br$ and $CH_3Cl$ emissions may be even higher under natural conditions.

**$CH_3Br$ and $CH_3Cl$ production in seawater**. Natural seawater from San Francisco Bay produced relatively small amounts of $CH_3Br$ and $CH_3Cl$ under dark or light conditions, in comparison to the Cu(II) addition treatments. The treatment with both Cu(II) amendment and ambient sunlight exposure showed the greatest positive average fluxes of both $CH_3Br$ and $CH_3Cl$ (Fig. 4), while samples undergoing other treatments without this combination displayed relatively smaller fluxes. Similar to the results of soils and model compound incubations, the large production of $CH_3Br$ and $CH_3Cl$ was contingent upon the presence of both Cu(II) and sunlight. Similarly, a cruise study in the Atlantic Ocean also identified a photochemical production pathway for methyl iodide ($CH_3I$) production under ambient sunlight conditions[30].

Many discharge routes of anthropogenic copper to the oceans exist, such as copper leaching from boat antifouling paint and aquaculture nets, copper residue from automobile brake pads in urban runoff, and atmospheric aerosol deposition[33,34]. Therefore, this study also suggests a previously unaccounted for, primarily anthropogenically driven $CH_3Br$ and $CH_3Cl$ source from seawater through the interaction of copper, sunlight, and dissolved organic matter.

**Potential mechanism of the observed phenomenon**. It has been well established that Fe(III) can foster the reaction between guaiacol and chloride ions to produce $CH_3Cl$[12]; during this process, the only byproduct of this reaction was 1,2-benzoquinone and the aromatic ring remained intact (Fig. 5). To test if Cu(II) can also catalyze this reaction, the mixture of guaiacol, KCl, and Cu(II) was incubated, which generated $CH_3Cl$ at a higher production rate than the mixture with catechol (Supplementary Fig. 3b).

This may be attributed to the more readily liberated methyl group from the methoxy group of guaiacol, either through an

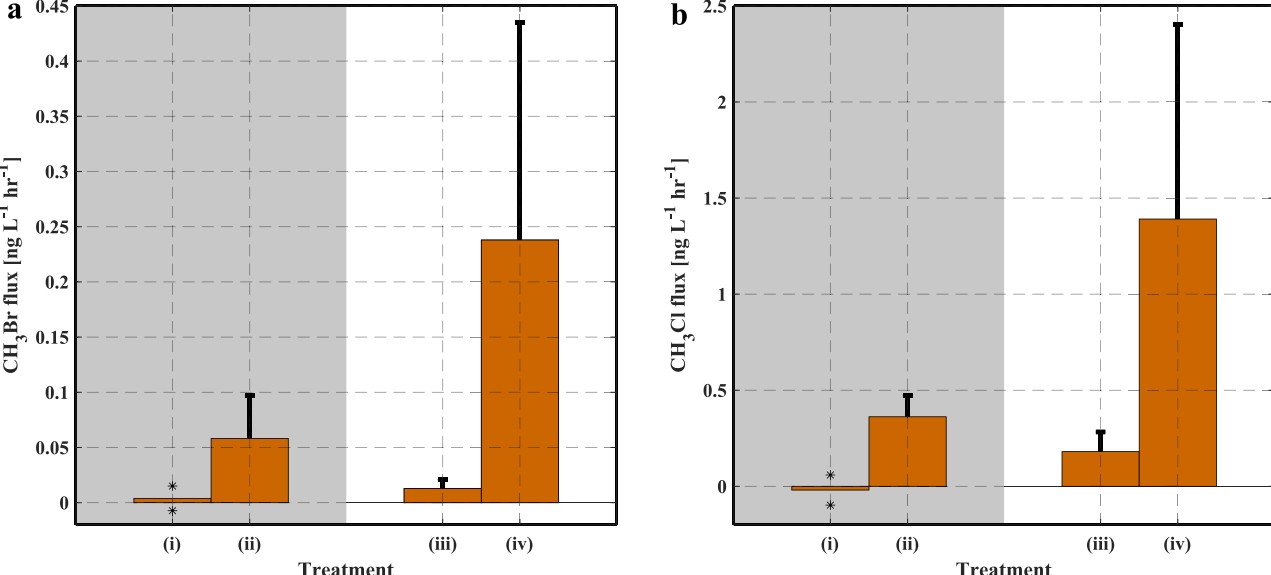

**Fig. 4 Methyl halide fluxes from seawater samples under different treatments. a** Methyl bromide ($CH_3Br$) and **b** methyl chloride ($CH_3Cl$) fluxes from the ocean water under four treatments. (i) seawater incubated under dark conditions ($n = 2$); (ii) seawater with Cu(II) amendment incubated under dark condition ($n = 4$); (iii) seawater incubated under ambient sunlight ($n = 5$); and (iv) seawater with Cu(II) amendment incubated under ambient sunlight ($n = 7$). The bars represent the average fluxes from replicate incubations of each treatment, and the error bars represent the standard deviations. For treatment (i), the two individual fluxes were presented in the figure (*), instead of the standard deviation.

**Fig. 5 Fe(III)-catalyzed[12] and presumed Cu(II)-catalyzed reaction pathway of guaiacol and halide ions to form methyl halides.** X represents Cl, Br, or I.

SN2 reaction (nucleophilic attack by $Cl^−$) or cleavage of $\cdot CH_3$ radical. The bond dissociation energy (BDE) required to cleave the $\cdot CH_3$ radical from the methoxy group of guaiacol is approximately 240 kJ/mol[35,36], the least within the structure. In comparison, the carbon bonds within the aromatic ring could possess up to 600 kJ/mol of BDE. Production of $CH_3Br$ and $CH_3Cl$ from catechol, which has no such methoxy group, must follow a different pathway, whereby the aromatic ring is fractured to provide $\cdot CH_3$ radicals.

Other studies showed that thermal decomposition of catechol and other aromatic compounds produces a range of $C_1-C_6$ products[22–24], suggesting the existence of thermal-facilitated ring-opening reactions. This could also potentially help explain the phenomenon that the same amount of Cu(II) addition would trigger a higher production of $CH_3Br$ and $CH_3Cl$ from the autoclaved soil than from the live soil. Hence, it is speculated that autoclaved soil produced non-aromatic carbon fragments, which reacted more readily in the production of halogenated volatile compounds.

Studies have identified multiple $C_1-C_6$ productions, such as furans[37,38], oxalic acid[39], carbon suboxide[40], chloroethane, iodoethane, and other chloroalkanes[21,41,42], chlorinated acetic acid[43,44] and other organic acids[39,45–47] (such as acetic acid, formic acid, fumaric acid, maleic acid, malonic acid, and muconic acid) as byproducts of the Fenton-like reactions of catechol[48]. We find this reaction also produces chloroform ($CHCl_3$) and $CO_2$ (Supplementary Table 4), compounds that have already been observed in previous similar studies[47,49]. Thus, it suggested that Fenton-like or photo-Fenton reactions also

resulted in multiple ring-opening reactions of catechol and produced radicals, such as $\cdot CH_3$, which facilitated the formation of halogenated organic compounds when halide ions were present (Supplementary Fig. 5).

**Environmental implications**. The mechanism as noted in this study, in which Cu(II) and radiation enhance the methylation of bromide and chloride to produce $CH_3Br$ and $CH_3Cl$, have been originally identified and used in bromide[20] and chloride[50] determination in aqueous samples at ultra-trace levels. This study presents a consideration of the environmental consequence of this process in both conventional and organic agriculture where large amounts of copper sulfate and other Cu(II)-based chemicals are applied, as well as in coastal marine environments where Cu(II)-containing runoff is discharged[51].

$CH_3Br$ and $CH_3Cl$ production induced by Cu(II)-based chemical usage may depend on several environmental variables, such as sunlight intensity, soil organic matter (content, composition, and structure), halide concentrations, etc. It has also been shown that pH of soil and water has an important influence on the occurrence of volatile halogenated compounds[31,43,49]. However, a simplified estimation without considering these factors suggests this process may be responsible for $4.1 \pm 1.9$ Gg $CH_3Br$ $yr^{−1}$ and $2.5 \pm 0.7$ Gg $CH_3Cl$ $yr^{−1}$, respectively (Supplementary Note 1). Similar to Fe(III)-induced production of methyl halides[12], this study indicates that application of copper(II)-based chemicals may increase atmospheric concentrations of methyl halides, especially for $CH_3Br$ (~10% of the missing sources), contribute to stratospheric halogen load, and thereby affect ozone levels.

## Methods

**Experimental procedure**. Three sets of experiments were conducted. The first set involved the incubation of soil samples with either unamended conditions or with chemical reagent supplements, such as copper(II) sulfate ($CuSO_4$) or hydrogen peroxide ($H_2O_2$) or both, and different light conditions, light vs. dark, to investigate their impact on $CH_3Br$ and $CH_3Cl$ production in soils. The second set of experiments involved incubations of solutions with catechol (benzene-1,2-diol) and guaiacol (2-methoxyphenol), compounds with (methoxy-) phenolic structures, which are often used as chemical models for natural organic matter in soil[12,37].

The third set involved incubation of seawater samples under different conditions: Cu(II) *vs.* no Cu(II) amendment, and light vs. dark. All three sets of incubations followed the same experimental set-up, as described in the section "Incubation and sampling", unless otherwise stated.

## Soil sample reactions

*Soil sample preparation.* Surface soil was collected from Oxford Tract (37°52′34″N 122°16′02″W), an agricultural research facility of the University of California, Berkeley. The soil type was a Tierra (Fine, montmorillonitic, thermic Molloc Palexerafls)—Urban land complex with a clay loam texture (soil properties in Supplementary Table 1). Before incubation, rocks and visible litter were removed, and the soil was sieved through a 2 mm standard test sieve (Fisher Scientific Company, New Hampshire, USA) to homogenize its texture. The soil was sealed and stored at about 5 °C and incubated within six months of collection. Part of the soil samples was oven-dried at 60 °C until constant weight to calculate soil water content based on the weight loss.

Autoclaved soil preparation (Supplementary Fig. 4): To differentiate between microbial and abiotic geochemical processes on $CH_3Br$ and $CH_3Cl$ production and degradation[12], a parallel set of soil samples was autoclaved at 105 °C overnight to sterilize the soil microorganisms and enzymes.

Mineral soil preparation (Supplementary Fig. 4): To test the necessity of soil organic matter, a parallel set of soil samples had their organic matter content depleted using the loss-on-ignition method: oven-dried soil samples were sieved through a 2 mm mesh and then heated in a muffle furnace (Thermolyne™, Thermo Fisher Scientific, Massachusetts, USA) at 375 °C for 24 hours to combust the soil organic matter.

*Incubation of soil samples.* Experiment I (results are presented in Fig. 1): To examine the effect of Cu(II) on $CH_3Br$ and $CH_3Cl$ production, live, autoclaved, and mineral soil samples (50 g each, dry weight, same hereafter) were mixed with 10 mL of solution containing $CuSO_4$ at 0 mM (deionized water), 1.25 mM, 2.5 mM, 5 mM or 10 mM. The glass jars containing all these mixtures were sealed and situated in a thermostatted bath (circulating water and ethylene glycol mixture, VWR Model 1180S, Pennsylvania, USA) at 20 °C during the experiments. Incubations to determine $CH_3Br$ and $CH_3Cl$ fluxes started within 30 min after the mixture preparation, and followed the method as described in the section "Incubation and sampling" below. Each treatment was conducted twice ($n = 2$).

Experiment II (results are presented in Fig. 2): To examine the combined effect of Cu(II) and $H_2O_2$ on $CH_3Br$ and $CH_3Cl$ production, live and autoclaved soil samples (50 g each) were incubated in glass jars under four different treatments. (i) soil samples were mixed with 20 ml of deionized water; (ii) soil sample was mixed with 10 ml of deionized water and 10 ml of $H_2O_2$ at 50 mM; (iii) soil sample was mixed with 10 ml of deionized water and 10 ml of $CuSO_4$ at 50 mM; (iv) 10 ml each of $H_2O_2$ at 50 mM and $CuSO_4$ at 50 mM were added into the soil samples. Incubations to determine $CH_3Br$ and $CH_3Cl$ fluxes started within 30 min after the mixture was completed, and followed the method described in the section "Incubation and sampling". Each treatment was conducted twice ($n = 2$).

Experiment III (results are presented in Fig. 3): To explore the persistence of the Cu(II) effect and to find out if a potential photochemical pathway involving ambient sunlight exists, a time-series light vs. dark experiment was conducted as follows.

Two groups of soil samples (50 g each) were mixed with 10 ml 50 mM $CuSO_4$ first in sealed glass jars and then incubated under dark and ambient sunlight conditions, respectively. Sunlight treatment was accomplished by exposing the glass jars under ambient sunlight on the roof of a building (10:00–17:00 PDT, July in 2021, 37°52′26″N 122°15′35″W). To account for the potential effect of temperature associated with sunlight exposure, the temperature within the light glass jars was measured in the beginning and the end of the experiment; then the dark glass jars were incubated at the same temperature as the immediate prior light experiments by situating in a temperature-controlled thermostatted bath (circulating water and ethylene glycol mixture, VWR Model 1180S, Pennsylvania, USA).

Prior to headspace air sampling, the glass jars were vented and flushed with ambient air for 1 min. After sealing the jars, 20 ml of headspace air was drawn into a glass syringe with Teflon plunger (SampleLok, Hamilton, Reno, Nevada, USA) and stainless steel needle through a septum on the cover of the glass jars at 30 min intervals for three samplings[52]. Sample storage time in the syringe was less than 5 min. The air samples were cryotrapped and then cryo-focused prior to injection into a gas chromatograph-mass spectrometer (Agilent 6890 N/5973, Agilent Technologies, California, USA) for $CH_3Br$ and $CH_3Cl$ analysis[8,53].

Each of these treatments was repeated three times ($n = 3$). The incubations for the dark experiment lasted for 8 days while the light experiment lasted for 17 days, until negative $CH_3Cl$ fluxes were finally observed. The solar radiation strength, such as, UVA ($\lambda = \sim 320–390$ nm), UVB ($\lambda = 265–322$ nm), and photosynthetic active radiation (PAR, $\lambda = 410–655$ nm), was also measured by UVA/UVB/PAR-BTA sensors (Vernier Inc., Beaverton, Oregon, USA) inside the glass jar in the sunlit experiments (Supplementary Table 2). Before the start of the experiment, parallel soil samples were mixed with 10 ml deionized water and incubated as controls.

## Model substance reactions

*Chemicals.* The following chemicals were used in the model substrate experiments: catechol, $C_6H_4(OH)_2$ (≥99%, Sigma-Aldrich); guaiacol, $C_6H_4(OH)(OCH_3)$ (≥99%, Sigma-Aldrich); potassium chloride, KCl (≥99.0%, Sigma-Aldrich),

potassium bromide, KBr (≥99.0%, Sigma-Aldrich); hydrogen peroxide, $H_2O_2$ (35%, Acros Organics, Thermo Fisher Scientific); copper(II) sulfate, $CuSO_4$ (≥99.99%, Sigma-Aldrich).

All the chemicals above were prepared as solutions with deionized water (Ultra-pure type I, ChemWorld, Utah, USA) and stored in volumetric flasks. Catechol and guaiacol were configured at 10 mM (millimole per liter; same hereafter); KBr and KCl were configured at 20 mM; $CuSO_4$ and $H_2O_2$ were configured at 50 mM.

*Incubation of model substances.* Experiment IV (results are presented in Supplementary Fig. 1): Control experiments were conducted with a mixture of 10 ml of 10 mM catechol, 20 ml water and 10 ml of 20 mM KBr (Br-control) for $CH_3Br$ studies or 20 mM KCl (Cl-control) for $CH_3Cl$ studies.

In order to investigate the response of $CH_3Br$ and $CH_3Cl$ formation to varied oxidizing environments, 10 ml of 50 mM $CuSO_4$ solution was first added to a set of parallel control mixtures (both Br-control and Cl-control), after which, 10 ml of $H_2O_2$ solution at 0 mM (deionized water), 10 mM, 20 mM, 30 mM, 40 mM or 50 mM were added into the mixture for incubation.

In order to investigate the response of $CH_3Br$ and $CH_3Cl$ formation to varied amount of Cu(II) addition, 10 ml of 50 mM $H_2O_2$ was first added to a set of parallel control mixtures (both Br-control and Cl-control), after which, 10 ml of $CuSO_4$ at 0 mM, 10 mM, 20 mM, 30 mM, 40 mM or 50 mM were added into the mixture for incubation.

All the aforementioned mixtures were incubated to determine $CH_3Br$ and $CH_3Cl$ fluxes following the method as described in the section "Incubation and sampling" below. Each of the incubation treatments was conducted twice ($n = 2$).

Experiment V (results are presented in Supplementary Fig. 3):

*Effect of solar radiation*: The potential role of solar radiation was tested as follows: 10 ml each of 10 mM catechol and 20 mM KCl was mixed in a sealed glass jar. Subsequently, 10 ml of 50 mM $CuSO_4$ was added. The mixture was exposed under sunlight for three hours (12:00–15:00 PDT, 2019/10/23, 37°52′26″N 122°15′35″W, Supplementary Table 3) and then brought indoors to the laboratory for incubation to get $CH_3Cl$ fluxes. For comparison, the same mixture without sunlight exposure was also incubated.

*Reactions with guaiacol*: In the presence of Fe(III), the methoxy group of guaiacol can undergo nucleophilic attack by chloride to produce $CH_3Cl$, without needing sunlight or microbial mediation[12]. To determine if Cu(II) behaved similarly to Fe(III) in this known pathway, two additional sets of experiments were conducted with 10 ml each of either 10 mM guaiacol or 10 mM catechol, 20 mM KCl, and 50 mM $CuSO_4$.

All the aforementioned mixtures were incubated following the method as described in the section "Incubation and sampling" below, from which $CH_3Cl$ fluxes were determined. Each treatment was conducted twice ($n = 2$).

## Seawater sample reactions

*Seawater sample preparation.* Unfiltered seawater samples were collected from the San Francisco Bay (37°51′45″N 122°18′50″W) using glass jars ($V = 1.9$ L, Ball Corporation, California, USA). The jars were previously acid washed and then rinsed with deionized water, ethanol, and acetone to ensure purity. Samples were stored at 5 °C after collection, prior to the testing.

*Incubation of seawater.* Experiment VI (results are presented in Fig. 4): Seawater incubations consisted of 500 ml samples in glass jars, subjected to the following 4 treatments: dark control ($n = 2$), sunlight control ($n = 5$), dark Cu(II) addition ($n = 4$), and sunlight Cu(II) addition ($n = 7$). The dark control experiment entailed storing the sample unsealed for 1.5 h in a dark environment. The sunlight control experiment entailed storing the sample unsealed for 1.5 h under ambient light (noontime, sunny day). The Cu(II) experiments entailed the addition of 0.3 mg $CuSO_4$ to the seawater, which were mixed and dissolved in the solution, before situating under dark or light conditions. Immediately following the above treatments, all the seawater samples were sealed for a closed headspace and incubated following the method described in the section "Incubation and sampling".

## Incubations and halocarbon analysis

*Incubation and sampling.* For all incubations (except for experiment III), the soils, seawater, or chemical mixtures were first mixed well in glass jars and then situated in a thermostatted bath (circulating water and ethylene glycol mixture, VWR Model 1180S, Pennsylvania, USA) at 20 °C. The jars were sealed with a stainless steel lid and Viton gasket and connected to a previously-evacuated stainless-steel loop (14.5 mL), which served as a sampling volume. The sample valve is at the top of the jar while the samples are at the bottom. The initial headspace was filled with ambient air, either outdoor air or laboratory air. Headspace air samples from the jars were drawn into the sampling loop at 30 min intervals, and then drawn through a cryotrap, followed by desorption, cryofocusing, and injection into a gas chromatograph-mass spectrometer (Agilent 6890N/5973, Agilent Technologies, California, USA) for $CH_3Br$ and $CH_3Cl$ analysis[8,53].

*Flux calculation.* $CH_3Br$ and $CH_3Cl$ fluxes from the samples were calculated based on the hourly averaged change of headspace concentration multiplied by moles of headspace air[5], and normalized to the dry mass of the soil (ng $kg^{-1}$ $hr^{-1}$) or the

volume of the seawater (ng $L^{-1}$ $hr^{-1}$); or were reported as the total mass of $CH_3Br$ and $CH_3Cl$ produced in pure chemical experiments (ng). Positive fluxes represented production of $CH_3Br$ and $CH_3Cl$ while negative fluxes represented consumption of $CH_3Br$ and $CH_3Cl$ from the headspace air. At least two ($n \geq 2$) incubations were conducted for each treatment of the soils, seawater, and chemical mixtures.

*Calibration.* Weekly calibration curves were constructed using whole ambient air standards collected at Niwot Ridge, Colorado, and calibrated at the Global Monitoring Division Laboratory of the National Oceanographic and Atmospheric Administration. Each calibration curve was made up of 10–18 data points by trapping various volumes of standard gas to capture the full range of chromatogram peak areas observed in the samples. The daily drift of instrumental signals was also corrected using multiple daily runs of the same standard[8]. The averaged instrumental precisions for $CH_3Br$ and $CH_3Cl$ after applying drift corrections were 1.9% and 1.0%, respectively[8]. Using the precision estimates, ambient concentrations, and sample size, the minimum detectable fluxes from soils were about $\pm 0.02$ ng $kg^{-1}$ $hr^{-1}$ for $CH_3Br$ and $\pm 0.58$ ng $kg^{-1}$ $hr^{-1}$ for $CH_3Cl$.

## Data availability
The authors declare that the data generated or analyzed in this study are available within the paper and its supplementary information files. Source data for figures are provided at the Zenodo repository (https://doi.org/10.5281/zenodo.5706329).

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

## Acknowledgements

This work was funded by National Science Foundation (grant EAR-1530375 to R.C.R.) and University of California Natural Reserves (Carol Baird Award to Y.J.). The authors would like to thank Andrew Beecher and Flora Xu for assistance with soil sample preparation and some of the GC-MSD analysis; Christina Wistrom for coordinating field work at Oxford Tract; professors Timothy M. Bowles, Christopher J. Chang, Alex T. Chow, and Garrison Sposito for information and discussion. Publication made possible in part by support from the Berkeley Research Impact Initiative (BRII) sponsored by the UC Berkeley Library.

## Author contributions

Y.J. and R.C.R. conceived of this study with preliminary experiments conducted by J.Y.K. and J.V.; Y.J., W.Z. and J.Y.K. carried out the main experiments; W.Z., M.J.D. and R.C.R. provided essential insight in data interpretation; Y.J. wrote the manuscript with inputs from all authors.

## Competing interests

The authors declare no competing interests.
