## [Peer Review File · Nature Communications]

REVIEWER COMMENTS

Reviewer #1 (Remarks to the Author):

Based on recognized discrepancies in sources and sinks of ozone-depleting MeX in the atmosphere, researchers have been investigating alternative natural synthetic abiotic processes operative in soils and (sea)waters; this study further contributes on these premises, notably those for example cited in refs 11, 12, 18 and 20. Accounting for similar chemistry to iron, novel evidence is presented for the possibility that widespread usage of Cu-based fungicides as a source which should not be neglected in the calculation of MeX burdens released to the atmosphere. This study should be of significant general interest to readership in this field and can be recommended for consideration following some major revisions.

It was somewhat difficult to read this manuscript due to the diversity of experiments undertaken and the general paucity of their descriptive detail hinders transparency. Many of the ensuing comments likely stem from this. Moreover, significant attention needs to be drawn to a better rationale for the manner in which production fluxes have been estimated as this is a central point of this study. I have concerns regarding the approach to the calculations.

Navigation and undertaking quick calculations would be much easier if all units were consistent throughout the text and figures in this article. Presently, a mix of moles and grams is used; despite being SI units, it is cumbersome to navigate the calculations.

Line 88: Was a 50 g or 5 g subsample used for these experiments, this is not clear, yet impacts all calculations.

Lines 134-140: In examining the impact of solar radiation, why were no studies conducted on soil samples amended with Cu(II) and/or H₂O₂ ?

Line 141: This compound is not explicitly halogenated but rather undergoes nucleophilic attack by a halide in the presence of an Fe(III) catalyst to produce the methylhalide. Please restructure this sentence.

Line 154, section 2.4.2: The reader must assume that the glass jars in which these incubations were undertaken were sealed so as to present a closed headspace for periodic sampling...this is not stated ?

Lines 166-169: Are all experiments conducted in open glass jars: how are fluxes determined by headspace sampling....are air currents controlled, how close to the sample surface is the sampler?...this is somewhat confusing....

Line 174: Why was such a convoluted calibration process conducted for GC-MS measurements when direct calibration against precise standards could have been undertaken. Such "double calibration" increases analytical uncertainties and it is not known from this study how such uncertainties were propagated when calculating the analytical results. Certainly, data for MeBr shows large relative uncertainty; could they actually be larger if full error propagation is implemented ? Uncertainty should not simply reflect the standard deviation of replicate results unless it can be demonstrated that this is the largest contributor (by a factor of 3) to the overall budget.

Figure 1: With production rates being time dependent (as evident in Figure 2), what exactly are these bar graphs exhibiting....were the 30 min interval headspace samples quantitated and summed over the course of a day to derive these daily production rates. Unfortunately, I am unable to find enough detailed information on exactly how measurements were made and the data treated to yield the values that are presented. Moreover, without such details, how does one explain a negative rate of production in a control...clearly the concentration of headspace analyte has dropped below some earlier baseline value ? Where does this initial value come from at the start of the experiment to cause it to decrease over time? Do we take direction from Figure 2 wherein we see that in a control sample, before addition of any Cu(II), that there is an intrinsic production rate that can be measured? This needs more clarity in either the experimental section or in relation to the discussion.

Figure 1: In connection with the mechanism, what accounts for the 10-fold enhanced rate of production of MeBr compared to MeCl, despite the endogenous concentration of Cl in the soil being 100-fold greater (Table S1).

Figure 2: Interestingly, in this closed system the rate of production decreases with time (days), possibly, as the authors suggest, due to consumption of Cu(II) in its role as an oxidant of soil organic matter. It would appear that in these (presumably closed) vessels the resulting Cu(I) does not undergo aerial oxidation in a catalytic fashion to continue the reaction and is thus depleted. In the "real" world, is it assumed that this remains the case. Perhaps some discussion of this would be useful.

Figure 2: Please label "Day 9" on the axis. The data plausibly suggest that the decreasing daily rate of production may be the result of depletion of Cu(II), but how do the authors account for the

rather small response to the recharged system on Day 9 wherein the production rate is no more than 17 % of that on Day 1 upon addition of fresh Cu(II). Clearly, the concentration of Cl ion remaining in the test mass of soil is some 10^5 -fold in magnitude greater than can be accounted for by summing up the total amount of MeCl generated over the 9 days of the experiment (~ 25 nmol/kg \times 50 g soil (?) = ~ 1 nmol compared to the initial 150 ppm Cl content = 200 μ mol)...hence reaction appears effectively pseudo zero order with respect to Cl. What other variable could have changed to reduce the output on Day 9?

Line 233: Can authors check for depletion of Cu(II), perhaps by ion chromatography?

Figure 3 caption and elsewhere: Please indicate what the error bars in each figure signify...simply a standard deviation of 2 replicate analyses or a full uncertainty comprising all propagated errors.

Lines 253-255: The large impact evident of the presence of peroxide could alter calculated fluxes significantly....this is also evident in Figure 4. As noted, peroxide may interact directly with the halide ion to generate the free radical or may reoxidize the Cu(I) or change the structure of the organic fraction, all of which may alter the mechanism and rate of production of the MeX.

Noteworthy, is that despite adding peroxide, it can also be generated in soil samples in the presence of solar radiation (lines 309-310, ref 21 and 29). With this large effect in mind, why were fluxes calculated (text S1) based on results from Figure 3 which correspond to test samples not exposed to sunlight? Why were no experiments conducted with the test soils dosed with Cu(II) and exposed to sunlight? Calculations in text S1 assume a role for sunlight (lines 55-57), amounting to 0.5y yet do not use solar based experimental data.

Figure 6, caption: This is confusing to have different symbols related to different experimental conditions..., either replace i, ii or iii "identifiers" with the symbols, or label the lines in the graph as I, ii or iii.

Line 392: Note that this is not a novel mechanism that has been discovered, but rather a novel consideration of application of a known process to the consideration of the impact of Cu(II) usage. It may be interesting to note that the catalytic effect of Cu(II) has been used analytically for quantitative generation of MeCl for the ultratrace determination of chloride in aqueous samples [J. Hu, R.E. Sturgeon, K. Nadeau, X. Hou, C. Zheng and L. Yang, Copper ion assisted photochemical vapor generation of chlorine for its sensitive determination by SF-ICPMS, Anal. Chem., 90, 4112 - 4118 (2018)] as well as for bromine [R.M. de Oliveira, D.L. G. Borges, P. Grinberg, Z. Mester and R.E. Sturgeon, Copper-ion Assisted Photochemical Vapor Generation of Bromide and Bromate, accepted for publication, J. Anal. At. Spectrom., 2021]. Copies available if interested.

Line 398: This report has not studied the impact of temperature on the various reactions leading to MeX generation and this statement should be deleted.

Lines 400-403: Please see comments throughout as well as below concerning these calculations.

Table S1: it would be interesting to know the (readily available or leachable) endogenous concentrations of Cu and Fe in this test soil. Fe(III) has earlier been shown to be a significant participant in the oxidation of soil organic matter and production of methylhalides (ref 11) and may account for their generation in control samples.

Table S2: Please pay consider significant figures for reporting data. Can the authors calculate the equivalent irradiance for the PAR and report it (perhaps in addition to the units of μ mol $m^{-2} s^{-1}$) for ease of comparison with the other sources.

Line S61: The authors refer to conditions that generated 8.9 nmol/kg/d of MeBr, corresponding to case (III) of Figure 3 which, according to the caption information corresponds to a solution containing 10 ml of 50 mM CuSO₄, which is 0.005 mol of Cu(II), not 0.00025; this alters calculated fluxes by two-fold. Please also see earlier comments related to this calculation and comment accordingly. This full calculation assumes overall production is linearly related to the amount of Cu(II) used or amended to the soil. The authors have not clearly established whether some kind of stoichiometric role exists for Cu(II) or even if it is catalytic so it is not clear how the use of the global mass of Cu-fungicide directly enters into the calculation. Further, should this estimate not have been based on the data in Fig 2 showing the impact of a single dose (application) of Cu producing ~ 23 nmol/kg MeCl and 31 nmol/kg MeBr? The authors show the significance of Cu(II) and organic matter as a requirement for MeX production but subsequently conclude with calculations based on sunlight....there is also no clear role proposed for the significant impact of Cu(II) other than suggesting that it functions in some Fenton reagent role? Why then are data taken from the results of Figure 3 for this calculation which did not expose the sample to sunlight yet the impact of solar radiation was factored into the flux calculations? This is inconsistent and not clear and requires considerably more rational discussion.

Please consider my comments available for any public release; Ralph Sturgeon

Reviewer #2 (Remarks to the Author):

Volatile halogenated compounds such as anthropogenic emitted chlorofluorocarbons (CFCs) effectively deplete the levels of stratospheric ozone. However, anthropogenic emissions of CFCs have been strongly reduced since the Montreal protocol came into force. Consequently, volatile halogenated compounds released from natural sources, such as methyl chloride (CH₃Cl) and methyl bromide (CH₃Br) have become increasingly relevant in stratospheric ozone depletion. Understanding the origin and fate of methyl halides in the environment is important in order to better predict their contribution to future stratospheric ozone destruction.

The study by Jiao and co-authors describes copper(II) induced abiotic formation of CH₃Cl and CH₃Br from soil and seawater samples during laboratory incubation experiments. The authors propose several formation pathways to explain their results and finally the obtained emission rates were extrapolated to the global scale.

The reported results of this study are novel, highly interesting and important as they further improve our understanding of natural but also anthropogenically induced formation of methyl halides. However, to improve the work substantial revision (mainly methodological, technical and presentation issues) is necessary. I have made comments listed below which the authors should pay attention to.

Main comments:

“The light issue”:

The authors propose in the abstract that solar radiation plays a role in forming halomethanes in soil and aquatic samples. In the main text of the manuscript, they show results of soil and water samples that have been exposed to solar radiation using transparent glass jars (line 136). What does this exactly mean? In Table S2 of the SI it gets clear that only a fraction of UV-B (about 24%) and PAR (about 77%) was reaching the sample inside the vial. When dealing with the role of UV light the authors refer to the manuscript by Yang et al. 2020 and the formation of CH₃Cl during irradiation of seawater by UV light. The results reported by Yang et al. (2020) show the application of artificial ultraviolet irradiation ($\lambda = 254$ nm) that does luckily not reach the surface of the Earth. Therefore, the pathway proposed by Yang et al. might be plausible in sewage treatment industries only, instead of natural environments. Instead, solely refer to Moore (2008) when discussing the natural effects of UV light on the formation of CH₃Cl in seawater. Moreover, I suggest revising all parts in the manuscript that deal with formation of methyl halides by UV radiation and clarify/specify this issue throughout other parts of the manuscript (in particular section 3.3). Remove UV light from scheme I and reference to Yang et al. (2020) as this work is not relevant for formation of methyl halides in natural terrestrial and aquatic environments. Make clear that when dealing with photochemical induced reactions that you applied solar radiation (including wavelength UV-A/B and visible light) and based on your experiments you can distinguish which wavelengths are primarily responsible for methyl halide formation. Thus, when describing your light experiments it is best to use solar radiation instead of saying it is UV light. You might discuss the potential role of UV radiation but you have not performed experiments (with specific UV-lamps or cut-off filters) that verify your claims. Furthermore, be also aware that your light exposure experiments have changed temperature in your incubation vials (please see also comment below).

“Incubation temperatures of experiments/treatments”:

Temperature is an important variable that might also strongly control formation rates of methyl halides in soil and seawater. However, it is very hard to find any information about the incubation temperature both in the Materials & Methods section but also not in the main text and Figure legends. These data need to be added and further discussed particularly when comparing results with treatments exposing incubation vials to solar radiation. As we can learn from the information provided in the SI (Table S2) temperature inside the vials increased substantially when exposed to solar radiation. Please discuss this issue in more detail in the main manuscript and add information where it is missing.

“Concentrations of halides in soil and water and relationship to formation of methyl halides”:

In Table S1 of the SI important information regarding halide concentration in the investigated soil has been provided. However, this information is missing in the main part of the manuscript particular with regard for discussion of preferential formation of CH₃Br relative to CH₃Cl. What is the molar ratio between produced CH₃Cl and CH₃Br and in comparison to the molar halogen ratio in the soil. This can be done for both untreated (no chloride or bromide added) and with the incubations where halides has been supplemented to the soil. You might compare your results with previous investigations (see for example Keppler et al. 2000 & 2003 and Whiskerman et al. 2008).

“Reduction of Cu(II) to Cu(I)”:

As outlined in Scheme 1 the authors propose that Cu(II) is acting as an oxidant in a similar way as when applying Fe(III). In previous studies reduction of Fe(III) to Fe(II) has been shown during alkylation of halides. I wonder if the authors have attempted to measure the reduction of Cu(II) to Cu(I) to support their hypothesis.

"The effect of pH in soil and water samples":

The potential important influence of the pH value on the occurrence of methyl halides in soil (e.g. Keppler et al. 2003) and water samples should be mentioned in the results and discussion of the manuscript.

Specific comments

Introduction, page 3, line 35: Please note that a study by Bahlmann et. (2019) has recently argued that that emissions of CH₃Cl from tropical plants (previously estimated at ~2000 Gg yr⁻¹) might be much lower (670±200 Gg yr⁻¹) suggesting a large unknown CH₃Cl source of 1530±200 Gg yr⁻¹.

Page 9, line 172: How is it possible to show error bar in Figs. 1-6 if only two incubations were performed? Please see also comment regarding statistics below.

Page 19, Scheme 1: Remove UV light from scheme and reference to Yang et al. (2020) as this work is not relevant for natural terrestrial and aquatic environments (see comments above "light issue").

Page 20, lines 370-371: change "...the same as a previous study" to "compounds that have been already observed in previous studies".

Page 21, Scheme 2: Please also include reference to Comba et al. (2015). This work provides an excellent overview about our knowledge of abiotic iron-induced and iron-catalysed oxidation of organic substrates in the presence of halides. In addition, refer to the work by Keppler et al. (2003) instead of "...assumed" (line 390). These authors have already shown formation of volatile iodinated alkanes in soil and propose two possible reaction pathways for the chemical formation of alkyl iodides.

Global estimates, page 22, lines 401-406: I would suggest presenting the calculated global estimates more carefully by saying that this approach represent a simplified global estimate that does not take into account environmental controls/variables such as temperature, pH in soil and water, light intensity, etc..

Moreover, the estimated number of around 23 Gg yr⁻¹ for CH₃Cl emission represents not an important part of the missing global source. Just say that might represent a part of the missing source.

Statistics:

For results presented in Figures 1 to 6 only two replicates of each treatment were measured. How is it then possible to present error bars/standard deviation? To my knowledge at least three values are required to calculate the standard deviation although excel software allows you to already calculate SDs from n=2. If only two replicates were measured for each treatment then you should present the mean value of the two replicates (and no error bars!) but also show the two single measurement for each treatment. Please accordingly modify all Figures including legends.

Mentioned references:

Bahlmann, E., Keppler, F., Wittmer, J., Greule, M., Schöler, H.F., Seifert, R., Zetzsch, C., 2019. Evidence for a major missing source in the global chloromethane budget from stable carbon isotopes. *Atmos. Chem. Phys.* 19, 1703-1719.

Comba, P., Kerscher, M., Krause, T., Schöler, H.F., 2015. Iron-catalysed oxidation and halogenation of organic matter in nature. *Environmental Chemistry* 12, 381-395.

Keppler, F., Eiden, R., Niedan, V., Pracht, J., Scholer, H.F., 2000. Halocarbons produced by natural oxidation processes during degradation of organic matter. *Nature* 403, 298-301.

Keppler, F., Borchers, R., Elsner, P., Fahimi, I., Pracht, J., Scholer, H.F., 2003. Formation of volatile iodinated alkanes in soil: results from laboratory studies. *Chemosphere* 52, 477-483.
Moore, R.M., 2008. A photochemical source of methyl chloride in saline waters. *Environ. Sci. Technol.* 42, 1933-1937.

Yang, Q. et al., 2020. Methyl chloride produced during UV254 irradiation of saline water. *J. Hazard. Mater.* 384, 121263.

Wishkerman, A., Gebhardt, S., McRoberts, C.W., Hamilton, J.T.G., Williams, J., Keppler, F., 2008. Abiotic methyl bromide formation from vegetation, and its strong dependence on temperature. *Environ. Sci. Technol.* 42, 6837-6842.

Point-by-point responses to the comments from the reviewers

We appreciate the reviewers for their time and effort spent reviewing our manuscript. The points and concerns raised by the reviewers have been carefully considered and addressed with substantial revisions. We believe that the manuscript is much improved as a result.

The point-by-point responses are listed below with reviewers' comments in black and our response in blue. In our responses, the line numbers (Line xxx) refer to the "clean" copy of the revised manuscript unless otherwise stated.

*****Reviewer #1*****

Reviewer #1 (Remarks to the Author):

Based on recognized discrepancies in sources and sinks of ozone-depleting MeX in the atmosphere, researchers have been investigating alternative natural synthetic abiotic processes operative in soils and (sea)waters; this study further contributes on these premises, notably those for example cited in refs 11, 12, 18 and 20. Accounting for similar chemistry to iron, novel evidence is presented for the possibility that widespread usage of Cu-based fungicides as a source which should not be neglected in the calculation of MeX burdens released to the atmosphere. This study should be of significant general interest to readership in this field and can be recommended for consideration following some major revisions.

Response: We thank Reviewer #1 for the positive appraisal of our work in general, and evaluating it as being of significant general interest to readership in this field.

We also noted Reviewer #1 mentioned that major revisions is required before it can be recommended for consideration. Therefore, we have revised the manuscript by fully considering, addressing and/or incorporating each of the comments. Please refer to our detailed responses below.

It was somewhat difficult to read this manuscript due to the diversity of experiments undertaken and the general paucity of their descriptive detail hinders transparency. Many of the ensuing comments likely stem from this. *(to be continued)*

Response: We appreciate Reviewer #2 for raising this general issue. The editor expressed a similar concern of our presentation of a diversity of experiments, which distracted from the main message or storyline of the paper.

In the revised manuscript, we have focused on the results and conclusions related to copper(II) additions to natural soil/seawater, and moved the ancillary experiments, such as pure chemical incubations, to the SI, which are referenced in related discussion in the main text. We have also restructured and rewrote the methodology part to ensure all the details of the experimental setup is included, so that other researchers can replicate our experiments.

We believe the manuscript is much improved in this respect. We kindly direct Reviewer #1 to the revised manuscript to see the change. One thing to note is that "method" part is now placed after "results and discussion", following the format of the journal.

(Continuing from the previous comment) Moreover, significant attention needs to be drawn to a better rational for the manner in which production fluxes have been estimated as this is a central point of this study. I have concerns regarding the approach to the calculations.

Response: We appreciate Reviewer #1's insight on the calculation of time-dependent and condition-dependent net methyl halide production associated with CuSO₄ additions, which is a central point of this study.

In the process of addressing this (and following Reviewer #1's suggestion below in the specific comments), we designed and conducted an additional experiment: time-series light vs. dark experiment (experiment III in the manuscript), in which we incubated raw soil samples with Cu(II) amendment under ambient sunlight (and dark) conditions, to better replicate the natural scenario of copper(II) application in agriculture. Based upon the CH₃Br and CH₃Cl production rates from this experiment, we extrapolated it to the global scale with some clearly stated caveats.

We kindly direct Reviewer #1 to experiment III (Lines 318-343), Text S1 (Lines 79-104 of the SI) and related discussion (Lines 157-171 and Lines 259-268) to view the updated calculation. We also have detailed responses to this issue throughout Reviewer #1's specific comments below.

Navigation and undertaking quick calculations would be much easier if all units were consistent throughout the text and figures in this article. Presently, a mix of moles and grams is used; despite being SI units, it is cumbersome to navigate the calculations.

Response: Following Reviewer #1's suggestion, in the revised manuscript, we have kept a consistent unit for soil CH₃Br and CH₃Cl fluxes, ng kg⁻¹ hr⁻¹, which is the mass of CH₃Br and CH₃Cl (in ng) produced from per kg of soil per hour. For seawater incubations, the unit was consistent as ng m⁻² hr⁻¹, which is the mass of CH₃Br and CH₃Cl (in ng) produced from per m² of seawater exposed at the air-water interface per hour. For chemical incubations, we kept the unit "ng" consistently, which is the mass of CH₃Br and CH₃Cl (in ng) produced from the chemical mixtures.

We hope this helps the readers to navigate through the results of the different experiments for comparison and calculation.

Specific comments:

Line 88: Was a 50 g or 5 g subsample used for these experiments, this is not clear, yet impacts all calculations.

Response: The soil sample was 50 grams for each of the incubations. We have added this information and made it clearer for the readers (also quoted below).

Experiment I - *"To examine the effect of Cu(II) on CH₃Br and CH₃Cl production, live, autoclaved, and mineral soil samples (50 g each, dry weight, same hereafter) were mixed..."* (Lines 300-301)

Experiment II - *“To examine the combined effect of Cu(II) and H₂O₂ on CH₃Br and CH₃Cl production, live and autoclaved soil samples (50 g each) were incubated in glass jars under four different treatments.”* (Lines 309-310)

Experiment III - *“Two groups of soil samples (50 g each) were mixed with 10 mL 50 mM CuSO₄ first in glass jars and then incubated under dark and ambient sunlight conditions, respectively.”* (Lines 321-322)

Lines 134-140: In examining the impact of solar radiation, why were no studies conducted on soil samples amended with Cu(II) and/or H₂O₂ ?

Response: In the revised manuscript, we have designed and conducted a set of time series experiments with light vs. dark incubations of soil samples amended with Cu(II) (experiment III), which also assessed the persistence of the enhanced production rates.

The description of the experimental setup, results/discussion can be found in Lines 318-343 and 157-171, respectively.

Line 141: This compound is not explicitly halogenated but rather undergoes nucleophilic attack by a halide in the presence of an Fe(III) catalyst to produce the methyl halide. Please restructure this sentence.

Response: We thank Reviewer #1 for the correction. We have restructured the sentence as follows.

“In the presence of Fe(III), the methoxy group of guaiacol can undergo nucleophilic attack by chloride to produce CH₃Cl, without needing sunlight or microbial mediation (ref. 12)” (Lines 378-380)

Line 154, section 2.4.2: The reader must assume that the glass jars in which these incubations were undertaken were sealed so as to present a closed headspace for periodic sampling...this is not stated?

Response: Yes, the glass jars were sealed to present a closed headspace. And the fluxes were measured based upon periodic sampling of the headspace air. As suggested by Reviewer #1, this is now explicitly stated in Lines 407-420 to make it clear for the readers.

“...The jars were sealed with a stainless steel lid and Viton gasket and connected to a previously-evacuated stainless-steel loop (14.5 mL), which served as a sampling volume. The sample valve is at the top of the jar while the samples are at the bottom. The initial headspace was filled with ambient air, either outdoor air or laboratory air. Headspace air samples from the jars were drawn into the sampling loop at 30 min intervals, and then drawn through a cryotrap, followed by desorption, cryofocusing, and injection into a gas chromatograph-mass spectrometer (Agilent 6890N/5973, Agilent Technologies, California, U.S.) for CH₃Br and CH₃Cl analysis (ref. 8, 53).

Flux calculation. CH₃Br and CH₃Cl fluxes from the samples were calculated based on the hourly averaged change of headspace concentration multiplied by moles of headspace air⁵, and normalized to the dry mass of the soil (ng kg⁻¹ hr⁻¹) or the area of the seawater exposed at the air-water interface (ng m⁻² hr⁻¹); or the total mass of

CH₃Br and CH₃Cl produced in pure chemical experiments (ng). Positive fluxes represented production of CH₃Br and CH₃Cl while negative fluxes represented consumption of CH₃Br and CH₃Cl from the headspace air...

Lines 166-169: Are all experiments conducted in open glass jars: how are fluxes determined by headspace sampling.....are air currents controlled, how close to the sample surface is the sampler?...this is somewhat confusing....

Response: (also following our response to the previous comment). The jars were sealed during the experiments with air withdrawn at 30 minute intervals for sampling. The sample valve is at the top of the jar while the samples are at the bottom. The change of CH₃X concentrations in the headspace air was highly linear ($R^2 > 0.90$), suggesting that this relatively passive mixing was sufficient. For demonstration purposes, a picture of the experimental setup with the sample jar outside of the thermostatted bath is shown on the right side.

Line 174: Why was such a convoluted calibration process conducted for GC-MS measurements when direct calibration against precise standards could have been undertaken. Such “double calibration” increases analytical uncertainties and it is not known from this study how such uncertainties were propagated when calculating the analytical results. Certainly, data for MeBr shows large relative uncertainty; could they actually be larger if full error propagation is implemented? Uncertainty should not simply reflect the standard deviation of replicate results unless it can be demonstrated that this is the largest contributor (by a factor of 3) to the overall budget.

Response: Weekly calibrations were needed (1) to verify the linearity of the instrumental (mass spectrometer) response to the amount of analyte trapped over the entire signal range; (2) to ensure that the trapping process is 100% for all sample volumes added; and (3) to account for any minor contamination in the system (only relevant for CH₃Cl, which had a very small but non-negligible blank signal). Daily calibrations were needed to improve accuracy, accounting for the drift in detector response over time. Repeated daily standards also provided instrumental precision for that day, which are reported on average as being 1.9% for CH₃Br and 1.0% for CH₃Cl. As the reviewer deduced, the concentrations are essentially determined by direct comparison with the daily standards, with the calibration curves largely verifying that the instrument is functioning as designed. Using the precision estimates and ambient concentrations, we report the minimum detectable fluxes as being $\pm 0.02 \text{ ng kg}^{-1} \text{ hr}^{-1}$ for CH₃Br and $\pm 0.58 \text{ ng kg}^{-1} \text{ hr}^{-1}$ for CH₃Cl. We omit the error in individual flux measurements for sake of clarity, as these errors are typically much smaller than the standard deviation or range of fluxes associated with replicate incubations. This is now stated in the methodology section (Lines 427-432) and SI (Lines 97-99).

Figure 1: With production rates being time dependent (as evident in Figure 2), what exactly are these bar graphs exhibiting.....were the 30 min interval headspace samples quantitated and summed over the course of a day to derive these daily production rates. Unfortunately, I am

unable to find enough detailed information on exactly how measurements were made and the data treated to yield the values that are presented. *(to be continued)*

Response: For each day's flux in Figure 1, it was calculated as the concentration change over time based upon the 30 min interval headspace samples (three consecutive samples in ~1.5 hours), which was then normalized to a daily basis ($\text{nmol kg}^{-1} \text{d}^{-1}$). For example, the flux was calculated as $A [\text{nmol kg}^{-1} \text{min}^{-1}]$, and then converted to $1440A [\text{nmol kg}^{-1} \text{d}^{-1}]$.

After considering Reviewer #1's comment, we agree that summation over the course of a day may not well represent the actual production rates, given that these production rates are time dependent, and it may lead to the misimpression that the flux measurements were conducted over a 24 hour period. Considering the experimental time scale was about 1.5 hours, we normalized the flux to an hourly basis ($\text{ng kg}^{-1} \text{hr}^{-1}$), in order to better represent the emission behavior of the samples on the experimental time scale. We have made it clear in Lines 411-413.

Please refer to Figure 1 to see the change. This change also applies to Figure 2 and 3 in the revised manuscript.

(Continuing from the previous comment) Moreover, without such details, how does one explain a negative rate of production in a control...clearly the concentration of headspace analyte has dropped below some earlier baseline value? Where does this initial value come from at the start of the experiment to cause it to decrease over time? *(to be continued)*

Response: The negative fluxes observed represent net consumption of CH_3X from the headspace air (a.k.a, as Reviewer #1 said, concentration of headspace air dropped below earlier baseline values). The initial headspace air was the ambient air which has a background CH_3Br and CH_3Cl concentration of $\sim 6.0 \text{ppt}^a$ and $\sim 550 \text{ppt}^b$, respectively. We have added the following description for negative fluxes in the revised manuscript.

"The initial headspace was filled with ambient air, either outdoor air or laboratory air." (Line 410)

"Positive fluxes represented production of CH_3Br and CH_3Cl while negative fluxes represented consumption of CH_3Br and CH_3Cl from the headspace air." (Lines 419-420)

(Continuing from the previous comment) Do we take direction from Figure 2 wherein we see that in a control sample, before addition of any Cu(II) , that there is an intrinsic production rate that can be measured? This needs more clarity in either the experimental section or in relation to the discussion.

Response: This is right. The control sample has an intrinsic production or consumption rates that can be measured, before addition of any Cu(II) . Following Reviewer #1's suggestion, we have made it clear in the revised manuscript.

^a Atmospheric CH_3Br concentrations from Global Monitoring Laboratory of NOAA - <https://gml.noaa.gov/hats/gases/CH3Br.html>

^b Atmospheric CH_3Cl concentrations from Global Monitoring Laboratory of NOAA - <https://gml.noaa.gov/hats/gases/CH3Cl.html>

“Time-series fluxes of (a) methyl bromide (CH₃Br) and (b) methyl chloride (CH₃Cl) from light vs. dark incubations of live soil samples (50 grams). On day 1, 10 ml of 50 mM CuSO₄ was added to soil samples” (Lines 152-154)

“Before the start of the experiment, parallel soil samples were mixed with 10 ml deionized water and incubated as controls.” (Lines 342-343)

Last thing we would like to mention is that we have conducted a new similar experiment III (Figure 3) to replace the Figure 2 in the previous submission. A comparison between the previous experiment (previous Figure 2) and the new experiment (new Figure 3) is as follows.

The previous Figure 2 was about CH₃Br and CH₃Cl fluxes from time-series incubation of autoclaved soil amended with Cu(II) under room light conditions

vs.

The new Figure 3 is about CH₃Br and CH₃Cl fluxes from time-series incubation of raw soil amended with Cu(II) under both sunlit and dark conditions.

Figure 1: In connection with the mechanism, what accounts for the 10-fold enhanced rate of production of MeBr compared to MeCl, despite the endogenous concentration of Cl in the soil being 100-fold greater (Table S1).

Response: We consistently observed disproportionate production of CH₃Br over CH₃Cl from the soil samples, especially given soil Cl concentrations being >100-fold greater. This production mechanism apparently favors Br to Cl when both are presented in the soil. The other reviewer also added a similar comment, referring us to some papers on Fe(III)-catalyzed methylation, in which the reaction also favors bromide over chloride. One literature proposed that this is possibly due to relative electronegativities and polarizabilities of the halogens, I > Br > Cl (Oliveira et al., 2021). Following both of the reviewers' comments, we have added the following discussion in Lines 89-95.

“Notably, live soils produced 4-6 times of CH₃Cl than CH₃Br on molar basis (e.g., amended with 10ml of 5 mM or 10 mM CuSO₄), despite Cl⁻ being ~135 times more abundant than Br⁻ in the soils (Table S1). For autoclaved soil, CH₃Br production rate was even up to 15 times that of CH₃Cl (e.g., amended with 10 ml of 5 mM CuSO₄). These results suggested that Cu(II)-catalyzed methylation reaction favors bromide over chloride when both ions are present, possibly due to relative electronegativities and polarizabilities of the halogens (ref. 20 - Oliveira et al., 2021). This is a similar trend as found in the Fe(III)-catalyzed production of methyl halides (ref. 12,13,21 - Keppler et al. 2000 & 2003 and Whiskerman et al. 2008)”. (Lines 89-95)

Similar inclination was also observed from the seawater incubation (experiment VI). However, we did not observe this inclination in the pure chemical incubations (experiment IV). It may be because either KBr or KCl was added separately (not at the same time), which potentially avoided the competition between the methylation of Br and Cl.

References:

Keppler, F., Eiden, R., Niedan, V., Pracht, J. & Schöler, H. F. Halocarbons produced by natural oxidation processes during degradation of organic matter. *Nature* 403, 298–301 (2000).

- Wishkerman, A. et al. Abiotic methyl bromide formation from vegetation, and its strong dependence on temperature. *Environ. Sci. Technol.* 42, 6837–6842 (2008).
- Keppler, F. et al. Formation of volatile iodinated alkanes in soil: results from laboratory studies. *Chemosphere* 52, 477–483 (2003).
- Oliveira, R. M. de, G. Borges, D. L., Grinberg, P., Mester, Z. & E. Sturgeon, R. Copper-ion assisted photochemical vapor generation of bromide and bromate. *J. Anal. At. Spectrom.* 36, 1235–1243 (2021).

Figure 2: Interestingly, in this closed system the rate of production decreases with time (days), possibly, as the authors suggest, due to consumption of Cu(II) in its role as an oxidant of soil organic matter. It would appear that in these (presumably closed) vessels the resulting Cu(I) does not undergo aerial oxidation in a catalytic fashion to continue the reaction and is thus depleted. In the “real” world, is it assumed that this remains the case. Perhaps some discussion of this would be useful.

Response: In our new experiment III, we observed the same phenomenon again that the rate of CH₃X production decreases over time. We believe this is true: that in real world scenarios, copper may undergo aerial and/or photochemical oxidation to Cu(II), thereby restoring the oxidized copper to continue the reaction. This is also proposed in a literature (Oliveria et al., 2021, see the reactions below).

Sunlit incubations in experiment III also strongly support this reasoning, showing that under light conditions, the enhancement effect Cu(II) on CH₃X production was not only larger but also for a twice as long in comparison to dark conditions.

Following Reviewer #1’s suggestion, we have added the following discussion.

“The depletion of Cu(II) may contribute to the persistent decline of CH₃Br and CH₃Cl production. Therefore, under natural conditions where copper can undergo aerial oxidation (ref. 20), it may apply a longer enhancing effect on CH₃Br and CH₃Cl production.” (Lines 159-162)

“On the other hand, the glass jars used in the sunlight experiments attenuated UVB ($\lambda = 265\text{-}322\text{ nm}$), UVA ($\lambda = \sim 320\text{-}390\text{ nm}$), and photosynthetic active radiation (PAR, $\lambda = 410\text{-}655\text{ nm}$) by about 75%, <1%, and 23%, respectively (Table S3), suggesting that CH₃Br and CH₃Cl emissions may be even higher under natural conditions.” (Lines 188-191)

Figure 2: Please label “Day 9” on the axis. The data plausibly suggest that the decreasing daily rate of production may be the result of depletion of Cu(II), but how do the authors account for the rather small response to the recharged system on Day 9 wherein the production rate is no more than 17 % of that on Day 1 upon addition of fresh Cu(II). Clearly, the concentration of Cl ion remaining in the test mass of soil is some 10⁵-fold in magnitude greater than can be accounted for by summing up the total amount of MeCl generated over the 9 days of the experiment ($\sim 25\text{ nmol/kg} \times 50\text{ g soil} (?) = \sim 1\text{ nmol}$ compared to the initial 150 ppm Cl content = 200 μmol)...hence reaction appears effectively pseudo zero order with respect to Cl. What other variable could have changed to reduce the output on Day 9?

Response: As noted above, we have replaced the experiment in Figure 2 with a new and more complete set of experiments shown in Figure 3. As for the observation that CH₃Cl emissions did not fully rebound after the addition of a resupply of CuSO₄, whereas CH₃Br did, we do not have enough data to supply a tenable hypothesis. As the reviewer notes, it is not likely due to the loss of chloride from the system. It may be a reaction that binds copper with chloride, but not bromide. It may also be attributable to the depletion of readily available carbon substrates and radicals (small molecules, fragments, etc) that are more easily chlorinated than brominated. The competition between bromide and chloride methylation reactions under limited carbon radical sources may also result the relative smaller production of CH₃Cl.

Line 233: Can authors check for depletion of Cu(II), perhaps by ion chromatography?

Response: Unfortunately, we were unable to measure Cu(II) for these experiments, nor were we able to find a laboratory that could do it for us. This will need to be addressed in a future set of experiments, as noted in the SI (Lines 56-58).

Figure 3 caption and elsewhere: Please indicate what the error bars in each figure signify...simply a standard deviation of 2 replicate analyses or a full uncertainty comprising all propagated errors.

Response: The error bars represented the calculated standard deviations of the replicates. However, as noted by Reviewer #2, the standard deviation (S.D.) of two data points is less meaningful than reporting both values, thereby providing their range. We agree and have revised all the figures by presenting mean and the two individual points instead of mean ± S.D. In the captions, we have made it clearer how the data is presented. We kindly direct Reviewer #1 to the captions to view the change.

Lines 253-255: The large impact evident of the presence of peroxide could alter calculated fluxes significantly....this is also evident in Figure 4. As noted, peroxide may interact directly with the halide ion to generate the free radical or may reoxidize the Cu(I) or change the structure of the organic fraction, all of which may alter the mechanism and rate of production of the MeX. Noteworthy, is that despite adding peroxide, it can also be generated in soil samples in the presence of solar radiation (lines 309-310, ref. 21 and 29). With this large effect in mind, why were fluxes calculated (text S1) based on results from Figure 3 which correspond to test samples not exposed to sunlight? Why were no experiments conducted with the test soils dosed with Cu(II) and exposed to sunlight? Calculations in text S1 assume a role for sunlight (lines 55-57), amounting to 0.5y yet do not use solar based experimental data.

Response: Following Reviewer #1's suggestion, we have designed and conducted a new experiment III, upon which the global extrapolation was based. To better simulate natural conditions of Cu(II) application on agricultural soils, experiment III was a time-series light vs. dark incubations, in which soil samples were amended with Cu(II) and then incubated with and without sunlight exposure, respectively. The detailed set up of this experiment can be found in Lines 318-343, and it is also quoted below in brief.

“...Two groups of soil samples (50 g each) were mixed with 10 ml 50 mM CuSO₄ first in glass jars and then incubated under dark and ambient sunlight conditions, respectively. Sunlight treatment was accomplished by exposing the glass jars under ambient sunlight on the roof of a building (10:00-17:00 PDT, July in 2021, 37°52'26"N 122°15'35"W). To account for the potential effect of temperature

associated with sunlight exposure, the temperature within the light glass jars was measured in the beginning and the end of the experiment; then the dark glass jars were incubated at the same temperature as the immediate prior light experiments by situating in a temperature-controlled thermostatted bath....”

The results of this experiment is presented in Figure 3, which is also shown below.

Figure 3. Time-series fluxes of (a) methyl bromide (CH₃Br) and (b) methyl chloride (CH₃Cl) from light vs. dark incubations of live soil samples (50 grams). On day 1, 10 ml of 50 mM CuSO₄ was added to soil samples; the dark incubation lasted for 8 days while the light incubation was 17 days. The circles represent the average fluxes of triplicate incubations ($n = 3$) and the error bars represent the standard deviations.

Based upon this experiment (Figure 3), 5×10^{-4} mol (10ml, 50mM) of Cu(II) addition to 50 grams of raw soil samples yielded 1656 ± 752 ng CH₃Br and 1018 ± 273 ng CH₃Cl, respectively, during the 17 days of ambient light exposure. This number was achieved by the integral of the flux-time curve (Figure 3) and then subtracted background fluxes from control studies (Day 0).

If we assume (1) the enhancement efficiency of Cu(II) on methyl halide production lasts ~17 days, as observed in the light experiments, (2) the soil used in this study is representative of global agricultural soil in terms of organic matter (content, composition and structure), chloride and bromide contents and other geochemical properties, and (3) omit the error in individual flux measurements for sake of clarity, as these errors are typically much smaller than the standard deviation associated with replicate incubations, it is extrapolated that the application of Cu(II) chemicals in global agriculture produced 4.1 ± 1.9 Gg CH₃Br yr⁻¹ (~10% of the missing sources) and 2.5 ± 0.7 Gg CH₃Cl yr⁻¹, respectively.

On the other hand, there are numerous caveats/flaws associated with simplified extrapolation (Text S1). Therefore, we present this extrapolation result with caution by explicitly mentioning that this is a simplified calculation, aiming to give a general picture on the strength of this possible mechanism. We also added the following text accompanying the extrapolation.

“CH₃Br and CH₃Cl production induced by Cu(II)-based chemical usage may depend on several environmental variables, such as sunlight intensity, soil organic matter (content, composition and structure), halide concentrations, etc. It has also shown that pH of soil and water has important influence on the occurrence of volatile

halogenated compounds (ref. 31, 43, 49 - Keppler et al. 2003; Huber et al., 2009; Liu et al., 2020). However, a simplified estimation without considering these factors..." (Lines 259-263)

References:

- Huber, S. G., Kotte, K., Schöler, H. F. & Williams, J. Natural abiotic formation of trihalomethanes in soil: results from laboratory studies and field samples. *Environ. Sci. Technol.* 43, 4934–4939 (2009).
- Liu, H. et al. Photochemical generation of methyl chloride from humic acid: impacts of precursor concentration, solution pH, solution salinity and ferric ion. *Int. J. Environ. Res. Public Health* 17, 503 (2020).
- Keppler, F. et al. Formation of volatile iodinated alkanes in soil: results from laboratory studies. *Chemosphere* 52, 477–483 (2003).

Figure 6, caption: This is confusing to have different symbols related to different experimental conditions..., either replace i, ii or iii “identifiers” with the symbols, or label the lines in the graph as I, ii or iii.

Response: We thank Reviewer #2 for the suggestion. This figure has been modified accordingly to avoid confusion. This updated figure has been moved to the SI (Figure S3).

Line 392: Note that this is not a novel mechanism that has been discovered, but rather a novel consideration of application of a known process to the consideration of the impact of Cu(II) usage. It may be interesting to note that the catalytic effect of Cu(II) has been used analytically for quantitative generation of MeCl for the ultratrace determination of chloride in aqueous samples [J. Hu, R.E. Sturgeon, K. Nadeau, X. Hou, C. Zheng and L. Yang, Copper ion assisted photochemical vapor generation of chlorine for its sensitive determination by SF-ICPMS, *Anal. Chem.*, 90, 4112 - 4118 (2018)] as well as for bromine [R.M. de Oliveira, D.L. G. Borges, P. Grinberg, Z. Mester and R.E. Sturgeon, Copper-ion Assisted Photochemical Vapor Generation of Bromide and Bromate, accepted for publication, *J. Anal. At. Spectrom.*, 2021]. Copies available if interested.

Response: We really appreciate the reviewer for letting us know of these two papers, which are closely related to the phenomenon we described in this manuscript. We did not realize that this phenomenon has been studied and even utilized in the chemical analysis of chloride and bromide at ultra-trace levels. Therefore, we deleted the statement “novel mechanism”, but directing the readers to the two publications, which elucidated relevant chemical mechanism and application of these reactions.

Following the Reviewer #1’s comments and information, we have rephrased the sentence as follows.

“The mechanism as noted in this study, in which Cu(II) and radiation enhance the methylation of bromide and chloride to produce CH₃Br and CH₃Cl, have been originally identified and used in bromide (ref. 20 - Oliveira et al., 2021) and chloride (ref. 50 - Hu et al., 2018) determination in aqueous samples at ultra-trace levels. This study presents a novel consideration of environmental consequence of this process in both conventional and organic agriculture where large amounts of copper sulfate and other Cu(II)-based chemicals are applied, as well as in coastal marine environments where Cu(II)-containing runoff is discharged (ref. 51)” (Lines 252-258)

We have also removed the phrase “novel mechanism” from the abstract. (Line 26)

The potential mechanism proposed in Oliveira et al (2021) also helped us elucidating the results.

“It has been proposed that the UVB region of sunlight may bring the complex of Cu(II) and halide ions to redox excited state (ref.20 - Oliveira et al., 2021), promoting methyl halide production. Therefore, the larger CH₃Br and CH₃Cl fluxes in samples with copper sulfate and sunlight implies that sunlight is an important driver for the reaction.” (Lines 184-187)

Line 398: This report has not studied the impact of temperature on the various reactions leading to MeX generation and this statement should be deleted.

Response: As suggested, we have deleted the temperature effect from the statement.

“CH₃Br and CH₃Cl production induced by Cu(II)-based chemical usage may depend on several environmental variables, such as sunlight intensity, soil organic matter (content, composition and structure), halide concentrations, etc.” (Lines 259-261)

Lines 400-403: Please see comments throughout as well as below concerning these calculations.

Response: We kindly direct Reviewer #1 to our comments above; Figure 3, Experiment III, and Text S1 in the revised manuscript to view the change.

Table S1: It would be interesting to know the (readily available or leachable) endogenous concentrations of Cu and Fe in this test soil. Fe(III) has earlier been shown to be a significant participant in the oxidation of soil organic matter and production of methyl halides (ref 11) and may account for their generation in control samples.

Response: We do not have this information at this time. From our communication with the long-time facilities manager, the soil at Oxford Tract has not been applied with Cu(II) or Fe(III) chemicals, neither has it been contaminated by such chemicals.

However, endogenous Fe and Cu may exist in the soils. For example, Fe(III) can originate as a mineral phase in the form of iron oxides and oxyhydroxides in the soil samples which could be reductively dissolved by natural organic compounds (Keppler et al., 2000; Lovley et al., 1991). Fe and Cu concentrations at Oxford Tract are probably at a similar magnitude as global uncontaminated soils. E.g., global soils have an extractable Fe concentration of about 0.45%-1.28% (Rossel et al., 2016).

Since the iron-induced production of methyl halide is ubiquitous (Keppler et al., 2000), we believe that the endogenous Fe (and Cu likely) contributed to the background CH₃X flux in control samples. We have added a short discussion in the main text -

“The deactivation of methyl halide-consuming microorganisms following heat treatment (Rhew et al., 2003) in autoclaved soil results in unidirectional interface of CH₃Cl and CH₃Br emissions, possibly catalyzed by endogenous Fe(III) in the soil (Keppler et al., 2000)” (Lines 73-76)

References:

- Keppler, F., Eiden, R., Niedan, V., Pracht, J., & Schöler, H. F. (2000). Halocarbons produced by natural oxidation processes during degradation of organic matter. *Nature*, 403(6767), 298-301.
- Lovley, D. R. (1991). Dissimilatory Fe(III) and Mn(IV) reduction. *Microbiological Reviews*, 55(2), 259-287.
- Rhew, R. C., Aydin, M. & Saltzman, E. S. Measuring terrestrial fluxes of methyl chloride and methyl bromide using a stable isotope tracer technique. *Geophys. Res. Lett.* 30, 2103 (2003).
- Rossel, R. V., Behrens, T., Ben-Dor, E., Brown, D. J., Demattê, J. A. M., Shepherd, K. D., ... & Ji, W. (2016). A global spectral library to characterize the world's soil. *Earth-Science Reviews*, 155, 198-230.

Table S2: Please pay consider significant figures for reporting data. Can the authors calculate the equivalent irradiance for the PAR and report it (perhaps in addition to the units of $\mu\text{mol m}^{-2} \text{s}^{-1}$) for ease of comparison with the other sources.

Response: Following Reviewer #1's suggestion, we have rounded the reported data (mean \pm standard deviation) based upon the resolution of measurements - our sensors (Vernier UVA/UVB/PAR-BTA) have a resolution of 5 mW m^{-2} , 0.25 mW m^{-2} and $1 \mu\text{mol m}^{-2} \text{ s}^{-1}$ for UVA, UVB and PAR, respectively. This update applies to both Table S2 and Table S3 (was Table S2 in the previous submission).

The PAR sensor reported measurements in terms of Photosynthetic Photon Flux Density ($\mu\text{mol m}^{-2} \text{ s}^{-1}$) calibrated for use in sunlight, but it did not provide a conversion factor to radiometric units (W m^{-2}). An approximation assuming a flat spectral distribution curve over 400-700 nm is $1 \text{ W m}^{-2} \approx 4.6 \mu\text{mol m}^{-2} \text{ s}^{-1}$ (https://www.licor.com/env/pdf/light/Rad_Meas.pdf). We have added this information in Line 66 of the SI and its footnote.

Line S61: The authors refer to conditions that generated 8.9 nmol/kg/d of MeBr, corresponding to case (III) of Figure 3 which, according to the caption information corresponds to a solution containing 10 ml of 50 mM CuSO_4 , which is 0.005 mol of Cu(II), not 0.00025; this alters calculated fluxes by two-fold. (*to be continued*)

Response: We appreciate Reviewer #2 for catching this important error. Calculations have been corrected and verified in the revised manuscript.

(Continuing from the previous comment) Please also see earlier comments related to this calculation and comment accordingly. This full calculation assumes overall production is linearly related to the amount of Cu(II) used or amended to the soil. The authors have not clearly established whether some kind of stoichiometric role exists for Cu(II) or even if it is catalytic so it is not clear how the use of the global mass of Cu-fungicide directly enters into the calculation. Further, should this estimate not have been based on the data in Fig 2 showing the impact of a single dose (application) of Cu producing $\sim 23 \text{ nmol/kg MeCl}$ and 31 nmol/kg MeBr ? The authors show the significance of Cu(II) and organic matter as a requirement for MeX production but subsequently conclude with calculations based on sunlight....there is also no clear role proposed for the significant impact of Cu(II) other than suggesting that it functions in some Fenton reagent role? Why then are data taken from the results of Figure 3 for this calculation which did not expose the sample to sunlight yet the impact of solar radiation was

factored into the flux calculations? This is inconsistent and not clear and requires considerably more rational discussion.

Response: We appreciate Reviewer #2 for the thorough review of our previous calculations. We agree that a sunlit exposed soil experiment may better replicate the natural scenario of Cu(II) chemical application in agriculture. Therefore, we have designed and conducted a new experiment (III) to revise our calculations, in which agricultural soil samples were amended with Cu(II) and incubated under both dark and sunlit treatment. We updated our calculation based upon this experiment.

We kindly direct Reviewer #1 to our response above; Figure 3, Experiment III, and Text S1 in the revised manuscript to view the change.

Please consider my comments available for any public release; Ralph Sturgeon

Response: We appreciate Dr. Ralph Sturgeon for his comments. We confirm that we have checked this option in the online submission system to release Dr. Ralph Sturgeon's comments and our responses.

*****Reviewer #2*****

Reviewer #2 (Remarks to the Author):

Volatile halogenated compounds such as anthropogenic emitted chlorofluorocarbons (CFCs) effectively deplete the levels of stratospheric ozone. However, anthropogenic emissions of CFCs have been strongly reduced since the Montreal protocol came into force. Consequently, volatile halogenated compounds released from natural sources, such as methyl chloride (CH₃Cl) and methyl bromide (CH₃Br) have become increasingly relevant in stratospheric ozone depletion. Understanding the origin and fate of methyl halides in the environment is important in order to better predict their contribution to future stratospheric ozone destruction.

The study by Jiao and co-authors describes copper(II) induced abiotic formation of CH₃Cl and CH₃Br from soil and seawater samples during laboratory incubation experiments. The authors propose several formation pathways to explain their results and finally the obtained emission rates.

The reported results of this study are novel, highly interesting and important as they further improve our understanding of natural but also anthropogenically induced formation of methyl halides.

Response: We thank Reviewer #2 for the assessing our work as novel, highly interesting and important for understanding the origin and fate of methyl halides in the environment in order to better predict their contribution to future stratospheric ozone destruction, and pointing out the environmental implications of this study, which is anthropogenic activities (in this case, it is Cu(II) chemical application) induced formation of methyl halides may contribute to atmospheric halogen loads.

However, to improve the work substantial revision (mainly methodological, technical and presentation issues) is necessary. I have made comments listed below which the authors should pay attention to.

Response: We acknowledge that Reviewer #2 mentioned that substantial revision is necessary. In accordance with comments, we have revised the manuscript by fully considering, addressing and/or incorporating all the comments and suggestions. Please refer to our detailed responses below.

Main comments:

“The light issue”:

The authors propose in the abstract that solar radiation plays a role in forming halomethanes in soil and aquatic samples. In the main text of the manuscript, they show results of soil and water samples that have been exposed to solar radiation using transparent glass jars (line 136). What does this exactly mean? In Table S2 of the SI it gets clear that only a fraction of UV-B (about 24%) and PAR (about 77%) was reaching the sample inside the vial. When dealing with the role of UV light the authors refer to the manuscript by Yang et al. 2020 and the formation of CH₃Cl during irradiation of seawater by UV light. The results reported by Yang et al. (2020) show the application of artificial ultraviolet irradiation ($\lambda = 254$ nm) that does luckily not reach the surface of the Earth. Therefore, the pathway proposed by Yang et al. might be plausible in sewage treatment industries only, instead of natural environments. Instead, solely refer to Moore (2008) when discussing the natural effects of UV light on the formation of CH₃Cl in seawater. *(to be continued)*

Response: We appreciate Reviewer #2 for raising this issue. The light treatment in our experiments was conducted by exposing the soil, chemical or seawater under ambient sunlight; while the study by Yang et al., (2020) was using artificial ultraviolet irradiation ($\lambda = 254$ nm), which does not occur in natural environments. In the revision, we now explicitly state this difference between Yang’s study and ours.

“For example, artificial ultraviolet irradiation ($\lambda = 254$ nm) can catalyze the reaction between dissolved organic carbon and chlorine in seawater to generate CH₃Cl (ref. 32 – Yang et al., 2020). However, in natural environments, UV radiation with such a short wavelength would have been absorbed by stratospheric ozone before it can reach earth’s surface, which makes this pathway only plausible in sewage treatment industries.” (Lines 179-183)

We retain references to other studies (including Moore, 2008) when discussing natural sunlight on the formation of CH₃Cl in seawater.

“Similar to some other studies (ref. 29, 30, 31 - Liu et al., 2020; Moore, 2008; Richter et al., 2004), the light involved in our experiments was ambient sunlight. It has been proposed that the UVB region of sunlight may bring the complex of Cu(II) and halide ions to redox excited state²⁰, promoting methyl halide production. Therefore, the larger CH₃Br and CH₃Cl fluxes in samples with copper sulfate and sunlight implies that sunlight is an important driver for the reaction.” (Lines 183-187)

“Similar to the results of soils and model compound incubations, CH₃Br and CH₃Cl production was contingent upon the presence of both Cu(II) and sunlight. Similarly,

a cruise study in the Atlantic Ocean also identified a photochemical production pathway for methyl iodide (CH₃I) production under ambient sunlight conditions (ref. 30 - Moore, 2008).” (Lines 197-201)

References:

- Liu, H. et al. Photochemical generation of methyl chloride from humic acid: impacts of precursor concentration, solution pH, solution salinity and ferric ion. *Int. J. Environ. Res. Public Health* 17, 503 (2020).
- Moore, R. M. A photochemical source of methyl chloride in saline waters. *Environ. Sci. Technol.* 42, 1933–1937 (2008).
- Richter, U. & Wallace, D. W. R. Production of methyl iodide in the tropical Atlantic Ocean. *Geophys. Res. Lett.* 31, (2004).

(Continuing from the previous comment) Moreover, I suggest revising all parts in the manuscript that deal with formation of methyl halides by UV radiation and clarify/specify this issue throughout other parts of the manuscript (in particular section 3.3). Remove UV light from scheme I and reference to Yang et al. (2020) as this work is not relevant for formation of methyl halides in natural terrestrial and aquatic environments. Make clear that when dealing with photochemical induced reactions that you applied solar radiation (including wavelength UV-A/B and visible light) and based on your experiments you can distinguish which wavelengths are primarily responsible for methyl halide formation. Thus, when describing your light experiments it is best to use solar radiation instead of saying it is UV light. You might discuss the potential role of UV radiation but you have not performed experiments (with specific UV-lamps or cut-off filters) that verify your claims. (to be continued)

Response: Following Reviewer #2’s suggestion, we have revised all parts in the manuscript to make it clear that our experiment showed solar radiation (wavelength including UVA, UVB and PAR) promoted the production of CH₃Br and CH₃Cl, instead of UV radiation alone, which was not separately tested and/or distinguished by our experiments. Scheme 1 is also revised as follows.

Scheme 1. Fe(III)-catalyzed (ref. 12 – Keppler et al., 2000) and presumed Cu(II)-catalyzed reaction pathway of guaiacol and halide ions to form methyl halides, X represents Cl, Br or I.

(Continuing from the previous comment) Furthermore, be also aware that your light exposure experiments have changed temperature in your incubation vials (please see also comment below).

Response: We kindly direct Reviewer #2 to our response below.

“Incubation temperatures of experiments/treatments”:

Temperature is an important variable that might also strongly control formation rates of methyl halides in soil and seawater. However, it is very hard to find any information about the incubation temperature both in the Materials & Methods section but also not in the main text and Figure legends. These data need to be added and further discussed particularly when

comparing results with treatments exposing incubation vials to solar radiation. As we can learn from the information provided in the SI (Table S2) temperature inside the vials increased substantially when exposed to solar radiation. Please discuss this issue in more detail in the main manuscript and add information where it is missing.

Response: We appreciate Reviewer #2 for raising this issue, as temperature and light conditions are usually associated with each other. In the revised manuscript, we clarify that all the incubations (except for the new experiment III) were conducted at 20 °C by situating the jars in a thermostatted bath (circulating water and ethylene glycol mixture, VWR Model 1180S, Pennsylvania, U.S.) at 20 °C. For the light vs. dark incubation of catechol (experiment V), the solution was exposed under ambient under sunlight for 3 hours and brought indoors to the lab for incubation. Therefore, the sunlit treated mixture was incubated at the same temperature as the dark experiment for CH₃Br and CH₃Cl fluxes.

For the new experiment III, to rule out the potential effect of temperature associated with sunlight exposure, temperature of the light glass jars was measured, and then the dark glass jars were incubated at the same temperature as the immediate prior light experiments by situating in a temperature-controlled thermostatted bath (circulating water and ethylene glycol mixture, VWR Model 1180S, Pennsylvania, U.S.).

Following Reviewer #2's concern, we have made this information clearer in the manuscript.

"...To account for the potential effect of temperature associated with sunlight exposure, the temperature within the light glass jars was measured in the beginning and the end of the experiment; then the dark glass jars were incubated at the same temperature as the immediate prior light experiments by situating in a temperature-controlled thermostatted bath (circulating water and ethylene glycol mixture, VWR Model 1180S, Pennsylvania, U.S.)..." (Lines 324-329)

"The potential role of solar radiation was tested as follows: 10ml each of 10 mM catechol and 20 mM KCl was mixed in a sealed glass jar. Subsequently, 10 ml of 50 mM CuSO₄ was added. The mixture was exposed under sunlight for three hours (12:00-15:00 PDT, 2019/10/23, 37°52'26"N 122°15'35"W, Table S3) and then brought indoors to the laboratory for incubation to get CH₃Cl fluxes..." (Lines 372-376)

"Concentrations of halides in soil and water and relationship to formation of methyl halides":

In Table S1 of the SI important information regarding halide concentration in the investigated soil has been provided. However, this information is missing in the main part of the manuscript particular with regard for discussion of preferential formation of CH₃Br relative to CH₃Cl. What is the molar ratio between produced CH₃Cl and CH₃Br and in comparison to the molar halogen ratio in the soil. This can be done for both untreated (no chloride or bromide added) and with the incubations where halides has been supplemented to the soil. You might compare your results with previous investigations (see for example Keppler et al. 2000 & 2003 and Whiskerman et al. 2008).

Response: We appreciate Reviewer #2 for raising this question. The other reviewer asked a similar question, which is a higher ratio of CH₃Br/CH₃Cl fluxes was observed in comparison to the ratio of Br/Cl contents in the soil samples. We found our results are consistent with the

conclusions from the references (Keppler et al. 2000 & 2003 and Whiskerman et al. 2008), which shows that soil halogen content is methylated by natural oxidation processes in the favored sequence: I > Br > Cl. This is possibly due to relative electronegativities and polarizabilities of the halogens (Oliveira et al., 2021). Following both reviewer's suggestion, we have calculated these ratios and added some discussion, quoted below.

“Notably, live soils produced 4-6 times of CH₃Cl than CH₃Br on molar basis (e.g., amended with 10ml of 5 mM or 10 mM CuSO₄), despite Cl⁻ being ~135 times more abundant than Br⁻ in the soils (Table S1). For autoclaved soil, CH₃Br production rate was even up to 15 times that of CH₃Cl (e.g., amended with 10 ml of 5 mM CuSO₄). These results suggested that Cu(II)-catalyzed methylation reaction favors bromide over chloride when both ions are present, possibly due to relative electronegativities and polarizabilities of the halogens (ref. 20 - Oliveira et al., 2021). This is a similar trend as found in the Fe(III)-catalyzed production of methyl halides (ref. 12, 13, 21 - Keppler et al. 2000 & 2003 and Whiskerman et al. 2008)”. (Lines 89-95)

References:

- Keppler, F., Eiden, R., Niedan, V., Pracht, J. & Schöler, H. F. Halocarbons produced by natural oxidation processes during degradation of organic matter. *Nature* 403, 298–301 (2000).
- Keppler, F. et al. Formation of volatile iodinated alkanes in soil: results from laboratory studies. *Chemosphere* 52, 477–483 (2003).
- Oliveira, R. M. de, G. Borges, D. L., Grinberg, P., Mester, Z. & E. Sturgeon, R. Copper-ion assisted photochemical vapor generation of bromide and bromate. *J. Anal. At. Spectrom.* 36, 1235–1243 (2021).
- Whiskerman, A. et al. Abiotic methyl bromide formation from vegetation, and its strong dependence on temperature. *Environ. Sci. Technol.* 42, 6837–6842 (2008).

“Reduction of Cu(II) to Cu(I)”:

As outlined in Scheme 1 the authors propose that Cu(II) is acting as an oxidant in a similar way as when applying Fe(III). In previous studies reduction of Fe(III) to Fe(II) has been shown during alkylation of halides. I wonder if the authors have attempted to measure the reduction of Cu(II) to Cu(I) to support their hypothesis.

Response: The measurement of copper transformation would be valuable to support the proposed mechanism, but this is outside the scope of the current study owing to our analytical limitations. Cu(I) ion itself is not soluble and is often bound with other ions to form intermediate complexes (Cuttell et al., 2002), complicating measurements. Following that, in the Scheme 1, we have stated that Cu(II)-catalyzed production of CH₃X is presumed. We kindly direct Reviewer #2 to the caption of Scheme 1 to view the change. We also add a note in Scheme S1 talking about this (Lines 56-58 of the SI).

Reference:

- Cuttell, D. G., Kuang, S. M., Fanwick, P. E., McMillin, D. R., & Walton, R. A. (2002). Simple Cu (I) complexes with unprecedented excited-state lifetimes. *J. Am. Chem. Soc.*, 124(1), 6-7.

“The effect of pH in soil and water samples”:

The potential important influence of the pH value on the occurrence of methyl halides in soil (e.g. Keppler et al. 2003) and water samples should be mentioned in the results and discussion of the manuscript.

Response: Though we did not include pH tests in our incubations, we find this paper and two other papers very relevant to our discussion on the influencing factors of CH₃Br and CH₃Cl fluxes. We've added a short discussion on the potential impact of pH on CH₃Br and CH₃Cl fluxes, quoted below.

“CH₃Br and CH₃Cl production induced by Cu(II)-based chemical usage may depend on several environmental variables, such as sunlight intensity, soil organic matter structures and contents, halide concentrations, etc. It has also shown that pH of soil and water has important influence on the occurrence of volatile halogenated compounds (ref. 31, 43, 49 - Keppler et al. 2003; Huber et al., 2009; Liu et al., 2020).”
(Lines 253-256)

References:

- Huber, S. G., Kotte, K., Schöler, H. F. & Williams, J. Natural abiotic formation of trihalomethanes in soil: results from laboratory studies and field samples. *Environ. Sci. Technol.* 43, 4934–4939 (2009).
- Liu, H. et al. Photochemical generation of methyl chloride from humic acid: impacts of precursor concentration, solution pH, solution salinity and ferric ion. *Int. J. Environ. Res. Public Health* 17, 503 (2020).
- Keppler, F. et al. Formation of volatile iodinated alkanes in soil: results from laboratory studies. *Chemosphere* 52, 477–483 (2003).

Specific comments:

Introduction, page 3, line 35: Please note that a study by Bahlmann et. (2019) has recently argued that that emissions of CH₃Cl from tropical plants (previously estimated at ~2000 Gg yr⁻¹) might be much lower (670±200 Gg yr⁻¹) suggesting a large unknown CH₃Cl source of 1530±200 Gg yr⁻¹.

Response: This publication provided updated data on the discrepancy between CH₃Cl sources and sinks. We have added this information in Lines 38-40, which is also quoted below.

“A recent reevaluation on CH₃Cl emissions from tropical plants, which was seen as the largest CH₃Cl source, showed it may be overestimated by about 1300 Gg yr⁻¹, suggesting the existence of even larger “missing” CH₃Cl sources than previously thought (ref. 2 - Bahlmann et al., 2019).” (Lines 38-40)

Page 9, line 172: How is it possible to show error bar in Figs. 1-6 if only two incubations were performed? Please see also comment regarding statistics below.

Response: After a careful consideration of Reviewer #2's comments, we agree that standard deviation of two data points doesn't convey information on the variation of data, other than the range of the two points (as shown in the equation below, \bar{x} is the mean of x_1 and x_2).

$$S. D. = \sqrt{\frac{(x_1 - \bar{x})^2 + (x_2 - \bar{x})^2}{2 - 1}} = \frac{\sqrt{2}}{2} |x_1 - x_2|$$

In the revised manuscript, we have followed the suggestion, and revised all the figures by presenting both of the two measurements, instead of mean ± standard deviation.

Page 19, Scheme 1: Remove UV light from scheme and reference to Yang et al. (2020) as this work is not relevant for natural terrestrial and aquatic environments (see comments above “light issue”).

Response: Done as suggested. We have deleted the “UV irradiation” from scheme 1. The new scheme is shown below.

Scheme 1. Fe(III)-catalyzed (ref. 12 – Keppler et al., 2000) and presumed Cu(II)-catalyzed reaction pathway of guaiacol and halide ions to form methyl halides, X represents Cl, Br or I.

Page 20, lines 370-371: change “...the same as a previous study” to “compounds that have been already observed in previous studies”.

Response: As suggested, we have changed “...the same as a previous study” to “compounds that have been already observed in previous...” (Lines 245-246)

Page 21, Scheme 2: Please also include reference to Comba et al. (2015). This work provides an excellent overview about our knowledge of abiotic iron-induced and iron-catalysed oxidation of organic substrates in the presence of halides. (*to be continued*)

Response: We thank Reviewer #2 for suggesting the reference, which is closely relevant to the discussion here. We have included this reference in Scheme S1 (which was scheme 2 in the previous submission).

(Continuing from the previous comment) In addition, refer to the work by Keppler et al. (2003) instead of “...(assumed)” (line 390). These authors have already shown formation of volatile iodinated alkanes in soil and propose two possible reaction pathways for the chemical formation of alkyl iodides.

Response: We thank Reviewer #2 for suggesting the reference. We deleted the word “assumed” for iodide in the scheme, and cited this paper instead.

Global estimates, page 22, lines 401-406: I would suggest presenting the calculated global estimates more carefully by saying that this approach represent a simplified global estimate that does not take into account environmental controls/variables such as temperature, pH in soil and water, light intensity, etc. (*to be continued*)

Response: We agree with this suggestion, since CH₃Br and CH₃Cl production induced by Cu(II)-based chemical usage may depend on several environmental variables, such as sunlight intensity, soil organic matter structures and contents, halide concentrations, etc. Following Reviewer #2’s suggestion, we explicitly stated in the revised manuscript that this is a simplified estimation, independent of potential environmental controls.

“CH₃Br and CH₃Cl production induced by Cu(II)-based chemical usage may depend on several environmental variables, such as sunlight intensity, soil organic matter (content, composition and structure), halide concentrations, etc. It has also shown that pH of soil and water has important influence on the occurrence of volatile halogenated compounds (ref. 31, 43, 49 - Keppler et al. 2003; Huber et al., 2009; Liu et al., 2020). However, a simplified estimation without considering these factors (calculation in Text S1) suggests this process may be responsible for 4.1 ± 1.9 Gg CH₃Br yr⁻¹ and 2.5 ± 0.7 Gg CH₃Cl yr⁻¹, respectively.” (Lines 259-265)

More discussion on the caveats of the extrapolation is given in Lines 91-98 of the SI.

(Continuing from the previous comment) Moreover, the estimated number of around 23 Gg yr⁻¹ for CH₃Cl emission represents not an important part of the missing global source. Just say that might represent a part of the missing source.

Response: The missing source of CH₃Cl is about 750 Gg yr⁻¹, therefore, this estimation suggested that it accounts for just a part of it. Accordingly, we have deleted the word “important” for CH₃Cl. For CH₃Br, our simplified estimation suggested that it may account for ~10% of the missing sources. We revised the text as follows.

“Similar to Fe(III)-induced production of methyl halides (ref. 12), this study indicates that application of copper(II)-based chemicals may increase atmospheric concentrations of methyl halides, especially for CH₃Br (~10% of the missing sources), contribute to stratospheric halogen load and thereby affect ozone levels.” (Lines 265-268)

Statistics: For results presented in Figures 1 to 6 only two replicates of each treatment were measured. How is it then possible to present error bars/standard deviation? To my knowledge at least three values are required to calculate the standard deviation although excel software allows you to already calculate SDs from n=2. If only two replicates were measured for each treatment then you should present the mean value of the two replicates (and no error bars!) but also show the two single measurement for each treatment. Please accordingly modify all Figures including legends.

Response: We have revised all the figures by presenting the data with mean and both of the two points, instead of mean ± standard deviation. We have also added this note in all the Figure captions. For the new experiment III, we conducted the incubations in triplicate (n = 3). Thus, we presented mean ± S.D. of fluxes in Figure 3, based on which, the global estimation was calculated.

Mentioned references:

Bahlmann, E., Keppler, F., Wittmer, J., Greule, M., Schöler, H.F., Seifert, R., Zetzsch, C., 2019. Evidence for a major missing source in the global chloromethane budget from stable carbon isotopes. *Atmos. Chem. Phys.* 19, 1703-1719.

Comba, P., Kerscher, M., Krause, T., Schöler, H.F., 2015. Iron-catalysed oxidation and halogenation of organic matter in nature. *Environmental Chemistry* 12, 381-395.

Keppler, F., Eiden, R., Niedan, V., Pracht, J., Scholer, H.F., 2000. Halocarbons produced by natural oxidation processes during degradation of organic matter. *Nature* 403, 298-301.

Keppler, F., Borchers, R., Elsner, P., Fahimi, I., Pracht, J., Scholer, H.F., 2003. Formation of volatile iodinated alkanes in soil: results from laboratory studies. *Chemosphere* 52, 477-483.
Moore, R.M., 2008. A photochemical source of methyl chloride in saline waters. *Environ. Sci. Technol.* 42, 1933-1937.

Yang, Q. et al., 2020. Methyl chloride produced during UV254 irradiation of saline water. *J. Hazard. Mater.* 384, 121263.

Wishkerman, A., Gebhardt, S., McRoberts, C.W., Hamilton, J.T.G., Williams, J., Keppler, F., 2008. Abiotic methyl bromide formation from vegetation, and its strong dependence on temperature. *Environ. Sci. Technol.* 42, 6837-6842.

Response: We appreciate Reviewer #2 for providing these references, which helped us elucidate the results of our study.

REVIEWERS' COMMENTS

Reviewer #1 (Remarks to the Author):

I have carefully read the replies to my earlier review comments and the manner in which they have been taken into account in revising the manuscript. I am satisfied with the actions taken to address my concerns and note that new experiments, restructuring, added details and extensive revisions have served to alleviate them. I find the authors have exercised due care during the review process and commented whenever a point was raised that they were simply unable to address due to logistical or other reasons. I find no additional actions are required. The paper remains as one of significant general interest to readers.

Only a few minor comments remain:

Main manuscript

Line 89: delete "of" change to "more"

Line 90: separate "10 ml"

Line 128: change "is" to "provides"

Line 220-221: delete "ahowed to produce" and insert "generated"

Line 303: separate mM from the numbers

Lines 360, 365 and 372: separate ml from the number

Supplementary Information file

Table S2: it is normal to report one more significant figure than is justified by its associated uncertainty, but this table of data reports too many justifiable significant figures...for example 1538 +/- 505 should be rounded to no more than 1540 +/- 510 as the uncertainty already starts in the 3rd decimal place. All numbers need to be corrected as well as that for the last line of Table S3

Reviewer #2 (Remarks to the Author):

The revised manuscript has substantially improved from previous version and most of my concerns have been satisfactorily addressed. However, there are a few issues (although minor) which should be addressed/clarified by the authors.

Specific comments:

- Scheme 1, page 11: For completion please add H+ on the product side of the reaction
- 2.6 Potential mechanism, page 11,12: The removal of the methyl group by the halide ion can be simply described as a SN2 reaction. However, following this paragraph the authors argue that methyl radicals are the intermediates to form methyl halides. This is somehow confusing. As both reaction types are possible for cleaving methyl carbon from methoxy group containing molecules and reacting with halides to form methyl halides, this should be made much clearer in the paragraph.
- section 2.5 CH3Br and CH3Cl production in seawater: I appreciate that the authors now present their results using consistent units. However, I wonder why they show the results of the seawater experiments in ng m⁻² hr⁻¹. This actually indicates a flux to the atmosphere whilst before (in the soil experiments) they provide the production rate on a weight basis (per kg). I cannot follow the reason for using m⁻² for the sweater experiments. I suggest to recalculate the seawater data and present results for example in ng l⁻¹ hr⁻¹

Point-by-point responses to the comments from the reviewers

*****Reviewer #1*****

Reviewer #1 (Remarks to the Author):

I have carefully read the replies to my earlier review comments and the manner in which they have been taken into account in revising the manuscript. I am satisfied with the actions taken to address my concerns and note that new experiments, restructuring, added details and extensive revisions have served to alleviate them. I find the authors have exercised due care during the review process and commented whenever a point was raised that they were simply unable to address due to logistical or other reasons. I find no additional actions are required. The paper remains as one of significant general interest to readers.

Response: We appreciate Reviewer #1 for the time and effort spent reviewing our manuscript for a second time.

Only a few minor comments remain:

Response: All the newly raised comments have been addressed and included in this revision. We kindly direct Reviewer#1 to our responses below and the revised manuscript to see the update.

Main manuscript

Line 89: delete “of” change to “more”

Response: Done as suggested.

Line 90: separate “10 ml”

Response: Done as suggested.

Line 128: change “is” to “provides”

Response: Done as suggested.

Line 220-221: delete “showed to produce” and insert “generated”

Response: Done as suggested.

Line 303: separate mM from the numbers

Response: Done as suggested.

Lines 360, 365 and 372: separate ml from the number

Response: Done as suggested.

Supplementary Information file

Table S2: it is normal to report one more significant figure than is justified by its associated uncertainty, but this table of data reports too many justifiable significant figures...for example 1538 +/- 505 should be rounded to no more than 1540 +/- 510 as the uncertainty already starts in the 3rd decimal place. All numbers need to be corrected as well as that for the last line of Table S3

Response: Done as suggested. We kindly direct Reviewer #1 to Supplementary Table S2 and S3 to see the change.

*****Reviewer #2*****

Reviewer #2 (Remarks to the Author):

The revised manuscript has substantially improved from previous version and most of my concerns have been satisfactorily addressed.

Response: We appreciate Reviewer #2 for the time and effort spent reviewing our manuscript for a second time.

However, there are a few issues (although minor) which should be addressed/clarified by the authors.

Response: All the newly raised comments have been addressed and incorporated in this revision. We kindly direct Reviewer#2 to our responses below and the revised manuscript to view the change.

Specific comments:

- Scheme 1, page 11: For completion please add H^+ on the product side of the reaction

Response: Following this suggestion, we have added H^+ on the product side of the reaction. We kindly direct Reviewer #2 to Fig. 5 (which was Scheme 1 in the previous version) to see the change.

- 2.6 Potential mechanism, page 11,12: The removal of the methyl group by the halide ion can be simply described as a SN_2 reaction. However, following this paragraph the authors argue that methyl radicals are the intermediates to form methyl halides. This is somehow confusing. As both reaction types are possible for cleaving methyl carbon from methoxy group containing molecules and reacting with halides to form methyl halides, this should be made much clearer in the paragraph.

Response: Following this suggestion, we now clarify that the methyl group can be liberated from the methoxy group either through SN_2 reactions or through cleavage of the methyl radical. We also deleted "methyl radicals are the intermediates to form methyl halides" in the following paragraph. The resulting paragraph (Lines 191-204) more clearly distinguishes the two reaction types.

"This may be attributed to the more readily liberated methyl group from the methoxy group of guaiacol, either through an SN_2 reaction (nucleophilic attack by Cl^-) or cleavage of $\cdot CH_3$ radical."

- section 2.5 CH_3Br and CH_3Cl production in seawater: I appreciate that the authors now present their results using consistent units. However, I wonder why they show the results of the seawater experiments in $ng\ m^{-2}\ hr^{-1}$. This actually indicates a flux to the atmosphere whilst before (in the soil experiments) they provide the production rate on a weight basis (per kg). I cannot follow the reason for using m^{-2} for the seawater experiments. I suggest to recalculate the seawater data and present results for example in $ng\ l^{-1}\ hr^{-1}$

Response: Following Reviewer #2's suggestion, we now normalize CH_3Br and CH_3Cl production rates to the volume of seawater, and report their production rates in the unit of $ng\ L^{-1}\ hr^{-1}$. We kindly direct Reviewer #2 to Fig. 4 and Line 389 to see the update.